# Conversation for Non-verifiable Learning: Self-Evolving Large Language Models through Meta-Evaluation

Yuan Sui [1] Bryan Hooi [1]

## Abstract

Training large language models (LLMs) for non-verifiable tasks—such as creative writing, dialogue, and ethical reasoning—remains challenging due to the absence of ground-truth labels. While LLM-as-Judge approaches offer a scalable alternative to human feedback, they face a fundamental limitation: performance is constrained by the evaluator's own quality. If the judge cannot recognize good solutions, it cannot provide useful training signals, and evaluation biases (e.g., favoring verbosity over quality) remain unaddressed. This motivates *meta-evaluation*—the ability to evaluate and improve the evaluator itself. We introduce CoNL, a framework that unifies generation, evaluation, and meta-evaluation through multi-agent self-play. Our key insight: *critique quality can be measured by whether it helps others improve their solutions*. In CoNL, multiple agents sharing the same policy engage in structured conversations to propose, critique, and revise solutions. Critiques that enable other agents' solution improvements earn a *diagnostic reward*, creating explicit supervision for meta-evaluation and enabling joint optimization of generation and judging capabilities through self-play, without external judges or ground truth. Experiments on various benchmarks show that CoNL achieves consistent improvements over self-rewarding baselines while maintaining stable training.[1]

## 1. Introduction

Large Language Models (LLMs) have demonstrated impressive performance on tasks with clear objectives, e.g., math, coding, and game-playing (Sui et al., 2025; Wang et al., 2025; Liao et al., 2025; Sui et al., 2026). However, many important applications involve *non-verifiable* tasks, such as creative writing, open-ended dialogue, and ethical decision-making, where objective ground-truth labels are absent (Bailey, 2025; Jia et al., 2025). In these settings, standard paradigms like Supervised Fine-Tuning (SFT) or Reinforcement Learning (RL) often break down due to the lack of verifiable signals.

To bridge this gap, *Reinforcement Learning from Human Feedback (RLHF)* (Christiano et al., 2017) has emerged as the standard solution, leveraging human preference judgments to guide model optimization. However, RLHF is expensive and does not scale well due to the high cost of human annotation. As an alternative, *LLM-as-Judge* methods have gained traction (Zheng et al., 2023; Gu et al., 2024), which employ LLMs to generate scalar rewards for scalable training. LLM-as-Judge approaches take two main forms: (1) Specialized Evaluators, such as Self-Taught Evaluators (Wang et al., 2024) and J1 (Whitehouse et al., 2025), which focus exclusively on training a separate judge model; and (2) Unified Models, such as Self-Rewarding LLMs (Yuan et al., 2024; Shafayat et al., 2025), where a single model acts as both generator and judge.

A fundamental problem remains across these approaches: the lack of *meta-evaluation*—evaluating the evaluators themselves. Current methods generally assume that evaluation capabilities will naturally improve as generation capabilities are trained, or rely on static external judges that cannot improve. Without scrutiny, models can exploit biases in the evaluation process. For instance, empirical studies show that self-rewarding models often exhibit a rapid increase in verbosity (e.g., response length jumping from 1k to 2.5k characters), suggesting the internal judge becomes biased toward longer, rather than better, responses (Wu et al., 2025a). Without a mechanism to evaluate the evaluator, these systems effectively operate in an echo chamber, capping performance at the model's initial bias level. This raises a question: can we train evaluation skills without ground truth?

We take inspiration from Wikipedia's peer review model. Wikipedia achieves high reliability not through a centralized oracle, but through collaborative improvement—contributors generate content, while others review

[1]National University of Singapore. Correspondence to: Yuan Sui <yuan.sui@u.nus.edu>.

[1]Code: https://github.com/Y-Sui/CoNL

and revise it. Quality emerges through this iterative refinement. Importantly, this process creates a natural proxy for evaluation quality: if a reviewer's feedback leads to improvements that the community accepts, that reviewer demonstrated good judgment. Similarly, we measure an agent's evaluation capability by whether its critiques enable other agents to produce better solutions—if a critique leads to measurable improvement, it identified real issues.

We formalize this insight in **CoNL** (**Co**nversation for **N**on-verifiable **L**earning), a multi-agent self-play framework that addresses the meta-evaluation challenge through a key insight: *the quality of a critique can be measured by whether it helps others to improve their solutions*. Consider a concrete scenario: Agent A proposes solution X, which the group initially rates moderately. Agent B then critiques X, identifying a logical flaw. Agent A revises the solution based on this feedback, fixing the error. The group then rates the revised solution higher than the original. This improvement reveals that Agent B's critique was constructive and identified a real issue. Crucially, this improvement signal can be measured and used as a training signal for meta-evaluation. Critiques that enable others to improve their solutions earn a *diagnostic reward*, creating explicit supervision for meta-evaluation without external judges or ground truth.

Our contributions are: (1) We introduce CoNL, a multi-agent self-play framework that unifies generation and meta-evaluation for non-verifiable tasks through conversation dynamics. (2) We propose measuring critique quality by whether it enables solution improvement, formalized as a *diagnostic reward* that tracks score changes after revision. (3) Experiments on diverse benchmarks show that CoNL outperforms self-rewarding baselines by 2.7-8.3 pp and closely matches RL with ground-truth rewards, while CoNL only uses peer consensus signals without ground-truth supervision. This demonstrates that meta-evaluation can be effectively trained through multi-agent conversations, opening new avenues for improving LLMs on non-verifiable tasks.

## 2. Related Work

**LLM-as-Judge and Meta-Evaluation.** The field has widely adopted LLM-as-Judge frameworks (Thakur et al., 2024; Li et al., 2024) to scale evaluation for non-verifiable tasks (Figure 1 contrasts the standard judge-based pipeline with our meta-evaluation setup). However, relying on a single judge model introduces a critical bottleneck: the evaluator's own biases (e.g., verbosity or position bias) place a hard ceiling on performance (Ye et al., 2025; Tan et al., 2025). While recent research acknowledges these flaws, it typically treats meta-evaluation as a post-hoc analysis rather than a training objective. This creates a problem: models are trained to generate, but their ability to judge remains static or implicitly assumed. CoNL addresses this

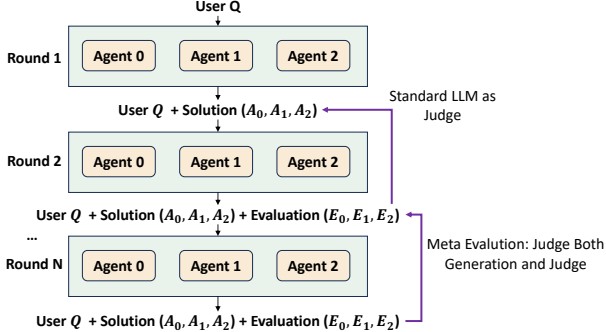

*Figure 1.* **Multi-Agent Conversational Paradigm with Meta-Evaluation.** Agents engage in iterative rounds where they generate solutions ($A$) and evaluate previous turns ($E$). Moving beyond standard *LLM-as-Judge*, the process incorporates *Meta-Evaluation* to scrutinize both the generation and the judging quality through peer consensus.

by treating meta-evaluation as a trainable capability. We use the success of a critique (its ability to improve other agents' solutions) as an explicit training signal, ensuring that evaluation skills improve alongside generation.

**Training Evaluators without Human Labels.** Recent efforts to train evaluators without human supervision fall into two main categories: self-improvement and verifiable RL. Self-improvement methods, such as Self-Taught Evaluators (Wang et al., 2024) and Meta-Rewarding LM (Wu et al., 2025b), rely on a model generating and judging its own synthetic data. However, this creates circular reasoning: a model's own judgments may reinforce its biases rather than correct them. Verifiable RL methods, like J1 (Whitehouse et al., 2025), solve this by grounding training in tasks with clear answers (like math), but this limits their applicability to domains where such answers exist. CoNL introduces a third paradigm: peer-supervision. By deriving signals from critique-driven improvement rather than self-judgment or ground truth, we avoid the circularity of self-training and the domain restrictions of verifiable RL.

**Critique-Driven Improvement.** A parallel line of work measures critique quality through downstream usefulness. RealCritic (Tang et al., 2025) evaluates critiques by their effect on revised answers but is evaluation-only and assumes ground-truth labels. RCO (Yu et al., 2025) trains a separate critic with a frozen 72B judge as supervision, while DeepSeekMath-V2 (Shao et al., 2025) bootstraps a verifier from expert annotations. CoNL shares the principle of grounding critique value in observable improvement, but differs in three ways: it requires no ground-truth labels, no external judge model, and trains a single policy that produces both solutions and critiques. Inference-time aggregation methods such as Compute as Teacher (Jayalath et al., 2025) and Recursive Self-Aggregation (Venkatraman et al., 2026) also use the model's own outputs as a stronger signal, but they aggregate at inference rather than producing a train-

ing signal. Our SRT-M baseline (majority voting within the four-round conversation) captures the strongest label-free version of this idea and is the most direct comparison for the contribution of $r_{\text{diag}}$.

**Multi-Agent Collaboration.** Multi-agent debate is a well-established strategy for improving inference performance (Du et al., 2024b; Zhang et al., 2025), where cross-examination filters out errors that a single agent might miss. However, existing frameworks view this interaction solely as a test-time cost, discarding the conversation data once the answer is produced. CoNL introduces a fundamentally different approach: we use critique-driven improvement as a training signal for meta-evaluation. Specifically, if an agent's critique enables another agent to improve their solution (measured by score increase $V_k^{\text{final}} > V_k^{\text{init}}$), that critique demonstrated good evaluation skills. This allows us to train not just generation quality, but also the ability to identify and diagnose genuine flaws—addressing the meta-evaluation problem through peer-supervised learning rather than external judges or ground truth.

## 3. Method

### 3.1. Problem Setting

We consider a learning problem where we want to train a model on tasks without verifiable solutions. In verifiable domains like mathematics and code generation, we can check if an answer is correct. In non-verifiable domains like creative writing, no such check exists. We have access only to human judgments $h : \mathcal{Y} \times \mathcal{X} \to \mathbb{R}$ (costly and often inconsistent) or LLM-based judges $J_\theta : \mathcal{Y} \times \mathcal{X} \to \mathbb{R}$ (scalable but unreliable). The unreliability of LLM judges stems from a fundamental limitation: they lack meta-evaluation—the ability to evaluate the quality of their own judgments. Without meta-evaluation, judges cannot distinguish between accurate and biased assessments, creating a performance ceiling.

CoNL addresses this by training a policy $\pi_\theta$ that unifies generation, evaluation, and meta-evaluation within a multi-agent self-play paradigm. We design a multi-agent conversation protocol where multiple agents sharing the same policy $\pi_\theta$ engage in multi-turn conversations to propose, critique, and revise solutions. Through these conversations, we extract training signals that teach $\pi_\theta$ not only to generate solutions, but also to evaluate whether its critiques correctly identify genuine flaws. The goal is to learn a policy that can generate high-quality solutions and accurately evaluate them, without relying on external judges or ground truth.

### 3.2. CoNL Protocol: Learning from Conversations

The key insight is that multi-agent conversations naturally reveal critique quality: if a critique enables another agent to improve their solution (measured by higher peer ratings

after revision), it likely identified a real issue. For each training query $q$, we instantiate $N$ agents from a single policy $\pi_\theta$. Each agent $i$ is assigned a distinct persona $P_i$ (e.g., "rigorous analyst", "creative solver", "skeptical reviewer"; see Appendix I) to encourage diverse perspectives and reduce the chance of group collusion (Wynn et al., 2025) (i.e., the tendency for group members to secretly cooperate for higher rewards). All agents share the same policy parameters $\theta$ but adopt different conversational roles. We design a four-round protocol as follows:

**Round 0: Initial Proposals.** Each agent $i$ independently generates an initial solution $s_i^{\text{init}}$ to query $q$.

**Round 1: Initial Evaluation and Critique.** After observing all initial solutions $\{s_j^{\text{init}}\}_{j=0}^{N-1}$, each agent $i$ produces:

- **Initial ranking** $\mathcal{R}_i^{\text{init}}$: A set of pairwise comparisons expressing preferences between solutions. For example, agent $i$ might state "Agent 2's solution is better than Agent 0's" and "Agent 1's solution is better than Agent 2's". We denote each comparison as a tuple $(a \succ b)$ indicating agent $a$'s solution is preferred over agent $b$'s. Crucially, agents do not see each other's rankings during this round, ensuring independent judgment.

- **Critiques** $\{c_{i \to k}\}_{k \in \mathcal{T}_i}$: textual critiques justifying the pairwise comparisons, where $\mathcal{T}_i$ denotes the set of agents mentioned in $\mathcal{R}_i^{\text{init}}$. Each critique $c_{i \to k}$ provides detailed reasoning for the comparisons involving agent $k$, identifying specific issues (e.g., logical errors, missing edge cases) or strengths in their solution.

**Round 2: Revision.** Each agent $i$ receives all critiques targeting them (i.e., all $c_{j \to i}$ where $i \in \mathcal{T}_j$) along with the original solutions. Agent $i$ then generates a revised solution $s_i^{\text{rev}}$, which may incorporate valid feedback and fix identified errors, defend against invalid or misguided critiques with justification, or refine reasoning and add missing details.

**Round 3: Final Verdict.** After observing all revised solutions $\{s_j^{\text{rev}}\}_{j=0}^{N-1}$, each agent $i$ produces a final ranking $\mathcal{R}_i^{\text{final}}$ via pairwise comparisons (similar format to Round 1). Aggregating these rankings yields post-conversation scores $V^{\text{final}}$, reflecting the group's updated assessment after critique and revision.

**Key design:** The initial ranking establishes a pre-conversation baseline $V^{\text{init}}$, which represents the group's initial assessment *before* agents see any critiques and before solutions are revised. This baseline helps measure whether the generated critiques lead to actual improvements after revision. Specifically, we measure the change in scores $\Delta V_k = V_k^{\text{final}} - V_k^{\text{init}}$ to quantify critique effectiveness. Agents do not see each other's rankings, ensuring independent judgment (detailed rationale in Appendix A). Figure 2 illustrates this four-round process; the full pseudocode is

provided in Algorithm 1.

Since multi-agent conversations can easily exceed the context window, we implement a memory buffering module that compresses past conversations while preserving key details (decisions, reasons, constraints). When the memory buffer is enabled, the output for each round is compressed into a condensed version for the next round's history (see Memory Buffer in Section 4.1 for implementation details).

### 3.3. Reward Design from Multi-Agent Conversations

After the four-round conversation, we have rich interaction data: initial solutions, initial rankings, critiques, revised solutions, and final rankings. The key question is: *how do we convert these informative conversations into numerical rewards that support model self-training?* Our approach is based on two core principles: (1) using pairwise comparisons as a robust evaluation mechanism, and (2) designing reward functions that explicitly measure different aspects of conversation quality.

**Pairwise Comparisons for Robust Evaluation.** As mentioned in Section 3.2, for initial ranking and final ranking, we prompt the agents to provide pairwise comparisons. Instead of providing absolute scores (e.g., "Agent 2's solution is 7/10"), we elicit pairwise preferences (e.g., "Agent 2's solution is better than Agent 0's"). This design choice has two key advantages: (1) Relative judgments are cognitively easier and more reliable than absolute scoring, and (2) Pairwise comparisons can naturally aggregate into quantitative scores like win rate.

To solve the potential conflicts in pairwise comparisons (e.g., Agent 0 prefers 1 over 2, but Agent 3 prefers 2 over 1), we use the Bradley-Terry (BT) (Bradley & Terry, 1952) model to aggregate them into latent quality scores (a self-contained walkthrough is provided in Appendix G). The BT model estimates each agent's latent "quality" $V_k \in [0, 1]$ by making it more likely that better agents win pairwise comparisons. Each agent produces multiple pairwise comparisons, which may conflict (e.g., agent 0 prefers 1 over 2, but agent 3 prefers 2 over 1). The BT model maximizes the likelihood that observed preferences match:

$$\mathbb{P}(\text{Agent } a \succ \text{Agent } b \mid V_a, V_b) = \frac{\exp(V_a)}{\exp(V_a) + \exp(V_b)}$$
(1)

Given all pairwise comparisons $\mathcal{R} = \bigcup_{i=0}^{N-1} \mathcal{R}_i$, we fit $\{V_k\}_{k=0}^{N-1}$ via maximum likelihood estimation. Higher $V_k$ means agent $k$'s solution is more preferred by the group.

We compute these aggregate quality scores at two timepoints: $V_k^{\text{init}}$ from initial rankings $\{\mathcal{R}_i^{\text{init}}\}_{i=0}^{N-1}$ captures the group's initial collective assessment (before seeing critiques and revisions), while $V_k^{\text{final}}$ from final rankings $\{\mathcal{R}_i^{\text{final}}\}_{i=0}^{N-1}$ captures the group's updated collective assessment after

critique and revision.

**Key insight:** The change $\Delta V_k = V_k^{\text{final}} - V_k^{\text{init}}$ measures whether agent $k$'s solution improved after revision. If agent $i$ critiqued $k$ and $\Delta V_k > 0$, it means agent $k$ was able to incorporate the feedback and produce a better solution, indicating that agent $i$'s critique correctly diagnosed real issues. This forms the basis for our diagnostic reward.

**Reward Functions from Conversation Dynamics.** From these conversation-derived quality scores, we design three reward components targeting different skills:

**Diagnostic Reward ($r_{\text{diag}}$).** We identify which agents were critiqued by agent $i$ by parsing the critique targets $\mathcal{T}_i$ from their Round 1 response (agents explicitly indicate critique targets in their output). If agent $k \in \mathcal{T}_i$ was critiqued and $k$'s revised solution receives a higher score ($V_k^{\text{final}} > V_k^{\text{init}}$), agent $i$ is rewarded:

$$r_{\text{diag}}(i) = \sum_{k \in \mathcal{T}_i} \max(0, V_k^{\text{final}} - V_k^{\text{init}})$$
(2)

The $\max(0, \cdot)$ ensures only diagnostic critiques (that correctly identify issues enabling improvement) receive positive reward; ineffective or misguided critiques will not be rewarded.

**Connection to meta-evaluation:** This reward directly addresses the fundamental challenge in LLM-as-Judge systems—how to evaluate the evaluators themselves. The diagnostic reward $r_{\text{diag}}$ provides a quantitative proxy for critique quality: if an agent's critiques consistently enable others to improve their solutions, that agent has strong evaluation skills. By training on $r_{\text{diag}}$, the model learns meta-evaluation—the ability to correctly diagnose genuine flaws and provide actionable feedback. This reward reflects both critique quality (did the critic identify real issues?) and revision quality (did the recipient successfully incorporate feedback?). Natural concerns about circular reasoning and false critique rewarding are addressed through bootstrapping and adversarial revision dynamics; we provide detailed analysis of these mechanisms in Appendix A.

**Solution Quality ($r_{\text{sol}}$).** Agents whose solutions receive high scores from the group after conversation receive higher rewards:

$$r_{\text{sol}}(i) = V_i^{\text{final}}$$
(3)

This directly rewards solutions that the group collectively ranks highly after the conversation.

**Majority Alignment ($r_{\text{meta}}$).** To prevent arbitrary or biased rankings, we reward agents whose pairwise judgments match the majority. Consider a pairwise comparison between agent $a$'s revised solution $s_a^{\text{rev}}$ and agent $b$'s revised solution $s_b^{\text{rev}}$. Let $\text{Pref}_i(a, b) \in \{a, b\}$ denote which solution agent $i$ prefers (e.g., if agent $i$ states "Agent $a$ is better than

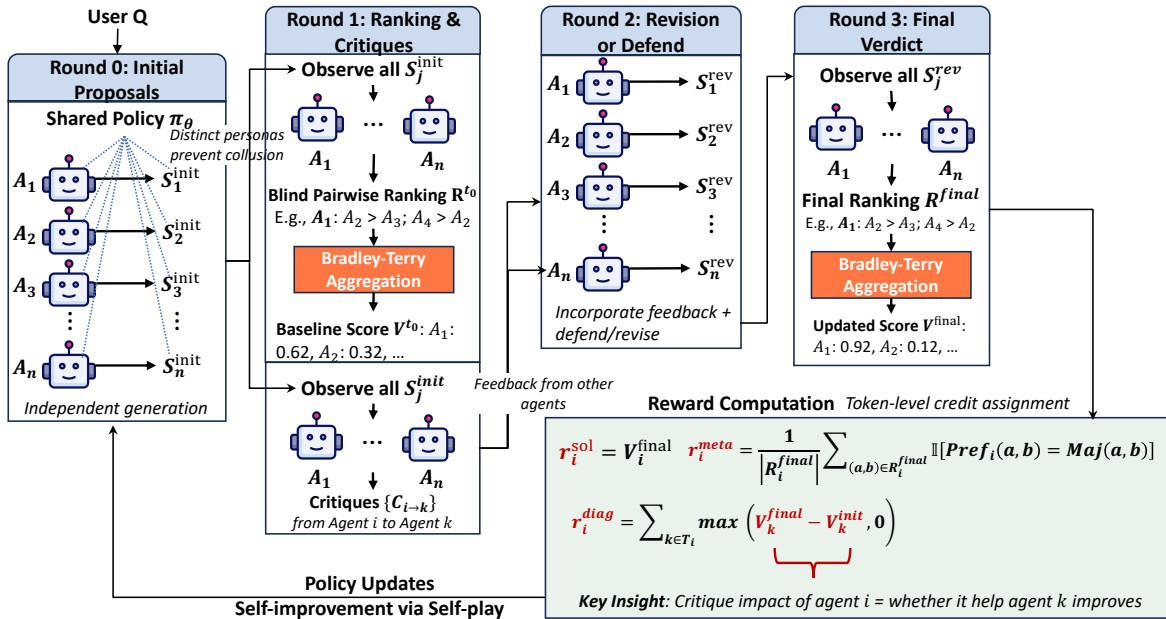

*Figure 2.* **Overview of CoNL training protocol.** Given a query, $N$ agents with diverse personas engage in a four-round conversation: (Round 0) each agent proposes an initial solution; (Round 1) agents provide initial rankings and critique peers; (Round 2) agents revise solutions based on received critiques; (Round 3) agents provide final rankings. We aggregate rankings via Bradley-Terry model to compute quality scores $V^{\text{init}}$ and $V^{\text{final}}$. Training rewards are computed from conversation dynamics: solution quality rewards better solutions, diagnostic rewards critiques that enable improvements ($V^{\text{final}} > V^{\text{init}}$), and consensus rewards rankings aligned with group majority.

*Table 1.* Credit assignment strategy. Each conversation segment receives targeted rewards. Critically, initial ranking tokens receive zero reward to prevent gaming the baseline.

| Round | Content | Reward | Rationale |
|---|---|---|---|
| 0 | Initial solution $s_i^{\text{init}}$ | $r_{\text{sol}}$ | Solution quality |
| 1 | Initial ranking $\mathcal{R}_i^{\text{init}}$ | **0 (masked)** | Prevent gaming $V^{\text{init}}$ |
| 1 | Critiques $\{c_{i \to k}\}$ | $r_{\text{diag}}$ | Diagnostic effectiveness |
| 2 | Revised solution $s_i^{\text{rev}}$ | $r_{\text{sol}}$ | Solution quality |
| 3 | Final ranking $\mathcal{R}_i^{\text{final}}$ | $r_{\text{meta}}$ | Majority alignment |

Agent $b$", then $\text{Pref}_i(a, b) = a$. We determine the majority preference $\text{Maj}(a, b) \in \{a, b\}$ by aggregating all agents' stated preferences on this pair. The majority-alignment reward is:

$$r_{\text{meta}}(i) = \frac{1}{|\mathcal{R}_i^{\text{final}}|} \sum_{(a,b) \in \mathcal{R}_i^{\text{final}}} \mathbb{I}[\text{Pref}_i(a, b) = \text{Maj}(a, b)]$$

(4)

This measures the fraction of agent $i$'s pairwise judgments that align with the group majority, ranging from 0 (no agreement) to 1 (perfect alignment).

The composite reward for agent $i$ is defined as:

$$r_{\text{total}}(i) = w_1 r_{\text{sol}}(i) + w_2 r_{\text{diag}}(i) + w_3 r_{\text{meta}}(i) \quad (5)$$

with default weights $w_1 = 1.0, w_2 = 2.0, w_3 = 1.0$. We weight diagnostic ($r_{\text{diag}}$) highest to emphasize meta-evaluation learning. The three rewards address complementary objectives: $r_{\text{sol}}$ trains generation quality, $r_{\text{diag}}$ trains

meta-evaluation, and $r_{\text{meta}}$ trains calibrated judgment. See Appendix A for detailed discussion of reward complementarity and diversity requirements.

### 3.4. Token-Level Credit Assignment

Each agent generates multiple text segments across rounds (initial solution, initial ranking, critiques, revised solution, final ranking). If we assign the same reward to all tokens, the model cannot distinguish which behavior led to which outcome. Instead, we perform segment-level credit assignment: each token receives the reward associated with its containing segment. In practice, we identify segment boundaries like [`<critique></critique>`] by parsing structured markers in the agent's output (implementation details in Appendix B). Each token within a segment receives the corresponding reward signal, enabling targeted gradient updates. See Table 1 for credit assignment.

### 3.5. Policy Training

We train $\pi_\theta$ using policy gradient methods implemented via the Tinker API (Thinking Machines AI, 2024). We employ importance sampling (IS) for unbiased gradient estimation when sampling and training distributions differ.

After collecting conversation trajectories from sampling policy $\pi_{\theta_{\text{old}}}$, we compute token-level advantages $\hat{A}_t$ based on segment-specific rewards (Table 1). The importance sampling objective corrects for distribution mismatch between

sampler and learner:

$$\mathcal{L}_{\text{IS}}(\theta) = \mathbb{E}_{x \sim \pi_{\theta_{\text{old}}}} \left[ \frac{p_\theta(x)}{p_{\theta_{\text{old}}}(x)} A(x) \right] \tag{6}$$

where the ratio $\frac{p_\theta(x)}{p_{\theta_{\text{old}}}(x)}$ reweights samples to account for policy updates. Complete mathematical formulations are provided in Appendix F.

## 4. Experiments

We evaluate CoNL on three non-verifiable benchmarks where ground-truth labels do not exist, and on five challenging verifiable benchmarks requiring advanced reasoning. Our experimental design addresses three questions: (1) Does CoNL improve performance on genuinely non-verifiable tasks without ground-truth supervision? (2) Can conversation-derived rewards effectively train both generation and evaluation skills without ground-truth labels? (3) On verifiable tasks, does the meta-evaluation signal ($r_{\text{diag}}$) outperform existing label-free baselines that use only peer agreement as the training signal?

*Table 2.* **Performance on non-verifiable benchmarks.** Values represent mean $\pm$ std over three independent runs. Methods marked with * require training. Otherwise, we report the inference performance. Ground-truth labels are never used by any method. See Table 9 for the per-category WildBench breakdown.

| Method | HealthBench | Research-Plan-Gen | WildBench |
|---|---|---|---|
| Base (0-shot) | $48.6 \pm 1.8$ | $16.2 \pm 1.5$ | 58.0 |
| Self-Consistency | $50.9 \pm 1.4$ | $18.4 \pm 1.3$ | 58.7 |
| Multi-Agent Debate | $52.3 \pm 1.6$ | $19.1 \pm 1.4$ | 59.4 |
| Self-Rewarding (SRT-M)* | $55.1 \pm 1.3$ | $22.7 \pm 1.2$ | 60.3 |
| **CoNL (Ours)*** | $\mathbf{59.2 \pm 1.1}$ | $\mathbf{27.9 \pm 0.9}$ | **64.1** |

### 4.1. Experimental Setup

**Datasets.** We conduct test-time training (He et al., 2025a) on diverse benchmarks spanning mathematics, graduate-level science, and competitive programming. Our training data consists of problems from DeepMath-103K (He et al., 2025b), AIME 2024 (Jia, 2024) and AIME 2025 (Zhang & Math-AI, 2025), GPQA Diamond (Rein et al., 2023), FrontierScience (Li et al., 2025), and USACO (Shi et al., 2024). From DeepMath-103K, which spans mathematical subjects including Algebra, Calculus, Number Theory, Geometry, Probability, and Discrete Mathematics, we randomly sample 3,500 questions with difficulty levels ranging from 6-10. We combine these with problems from the other benchmarks to create a diverse training distribution covering mathematics, graduate-level science, and competitive programming. We evaluate using Pass@1 accuracy (correctness of top-ranked solution), Pass@K (whether any top-K solutions are correct), and Rank-$\rho$ (Spearman correlation measuring evaluation quality). Detailed dataset and metric descriptions are in Sections C and E.

**Models and Infrastructure.** We train and evaluate using Tinker (Thinking Machines AI, 2024)—a scalable training platform for reinforcement learning with language models. We use four open-source models: Qwen/Qwen3-8B-Instruct and meta-llama/Llama-3.1-8B-Instruct as primary models, with Qwen/Qwen3-4B-Instruct-2507 and meta-llama/Llama-3.2-3B for smaller-scale experiments.

**Training Configuration.** We train with $N = 4$ agents assigned diverse personas (Appendix B) for the main results. We use importance sampling with learning rate $3 \times 10^{-5}$ and reward weights $w_1 = 1.0$ (quality), $w_2 = 2.0$ (diagnostic), $w_3 = 1.0$ (alignment). Crucially, ground-truth labels are never used for reward computation—all training signals derive purely from inter-agent comparisons.

**Memory Buffer.** The Tinker API constrains all models' context windows to 32k tokens. Since our multi-round multi-agent conversations can exceed this limit, we implement a memory buffering module to compress conversation history while preserving key details (see Appendix I for implementation details).

**Inference Budget.** All baselines and CoNL use a per-response budget of `max_tokens`=16,000. The Qwen3 technical report uses 32k+ tokens, which yields longer thinking chains and roughly 10 absolute points higher on AIME; since every method in our comparison uses the same budget, the relative gains across methods remain unaffected.

**Baseline Methods.** We consider baselines spanning inference-only and training-based methods. Inference-only baselines include: **Base (0-shot)** with Chain-of-Thought prompting, **Self-Consistency (SC)** (Wang et al., 2022), **Self-Refine (SR)** (Madaan et al., 2023), and **Multi-Agent Debate (MAD)** (Du et al., 2024a). Training-based baselines include two self-rewarding variants (Shafayat et al., 2025): **SRT-S (Single-turn)** uses majority voting on independent solutions as proxy ground truth, while **SRT-M (Multi-agent)** applies majority voting within our four-round conversation structure but without diagnostic rewards. Both SRT variants use the same training configuration as CoNL for fair comparison. See detailed descriptions in Appendix D.

### 4.2. Non-verifiable Tasks

We first evaluate CoNL on non-verifiable benchmarks where ground-truth labels do not exist, e.g., HealthBench (Arora et al., 2025), Research-Plan-Gen (Goel et al., 2025), and WildBench (Lin et al., 2025). These benchmarks match the motivating setting of CoNL: response quality is judged by humans or domain experts against subjective rubrics rather than against an objective ground truth.

Table 2 shows that CoNL outperforms SRT-M, the strongest label-free baseline, by a significant margin across all three benchmarks. These benchmarks score outputs against crite-

*Table 3.* **Pass@1 performance on verifiable reasoning benchmarks.** Values represent mean $\pm$ std over three independent runs. **Bold** and underline indicate the best and second-best performance within each model size category. Methods marked with * require training. Otherwise, we report the inference performance.

| Model | Method | AIME24 | AIME25 | GPQA | DeepMath | FrontierSci | USACO |
|---|---|---|---|---|---|---|---|
| **Qwen3-8B** | 0-Shot Inference | 60.0±1.8 | 60.0±2.1 | 67.0±0.9 | 70.5±0.6 | 43.0±1.4 | 10.2±0.8 |
| | Self-Consistency | 61.8±0.9 | 61.2±1.1 | 68.2±0.8 | 75.4±0.5 | 43.1±1.5 | 10.3±0.9 |
| | Self-Refine | 63.2±1.6 | 62.8±1.4 | 69.5±1.0 | 72.2±0.9 | 45.5±1.3 | 11.8±1.2 |
| | Multi-Agent Debate | 64.5±2.4 | 64.2±2.5 | 70.8±2.1 | 71.0±1.8 | 44.2±2.6 | 11.0±2.2 |
| | Self-Rewarding (Single Turn)* | 68.5±1.2 | 69.0±1.3 | 73.5±1.1 | 77.5±0.8 | 50.5±1.7 | 15.5±1.4 |
| | Self-Rewarding (Multi-Agent)* | 69.8±1.5 | 70.2±1.6 | 74.8±1.3 | 78.8±1.2 | 52.0±1.8 | 16.8±1.6 |
| | **CoNL (Ours)*** | **76.5±1.4** | **73.5±1.5** | **79.2±1.2** | **87.1±0.7** | **55.7±1.6** | **19.5±1.5** |
| **Qwen3-4B-Instruct** | 0-Shot Inference | 50.0±1.3 | 54.0±1.6 | 45.0±1.1 | 78.5±1.2 | 18.0±1.5 | 6.0±0.4 |
| | Self-Consistency | 51.5±0.8 | 55.2±1.0 | 46.1±0.9 | 79.8±0.7 | 17.1±1.4 | 6.1±0.5 |
| | Self-Refine | 52.8±1.4 | 56.8±1.5 | 47.5±1.2 | 80.9±1.3 | 19.2±1.8 | 6.9±1.1 |
| | Multi-Agent Debate | 54.1±2.1 | 57.9±2.3 | 48.8±1.9 | 81.5±1.6 | 19.9±2.4 | 6.5±1.9 |
| | Self-Rewarding (Single Turn)* | 57.5±1.3 | 61.2±1.4 | 51.5±1.2 | 83.8±1.1 | 23.5±1.7 | 9.5±1.3 |
| | Self-Rewarding (Multi-Agent)* | 58.9±1.5 | 62.5±1.6 | 52.8±1.4 | **85.2±1.4** | 24.8±1.9 | 10.2±1.5 |
| | **CoNL (Ours)** | **63.5±1.3** | **67.4±1.4** | **55.2±1.3** | 84.9±1.0 | **27.5±1.7** | **13.4±1.4** |
| **Llama-3.1-8B** | 0-Shot Inference | 13.0±0.8 | 7.0±0.4 | 23.0±1.0 | 45.0±1.5 | 1.9±0.2 | 3.0±0.3 |
| | Self-Consistency | 13.5±0.6 | 7.2±0.5 | 23.6±0.8 | 46.1±0.9 | 1.9±0.3 | 3.0±0.3 |
| | Self-Refine | 14.8±1.0 | 8.5±0.8 | 24.8±1.1 | 47.5±1.4 | 2.8±0.5 | 3.6±0.6 |
| | Multi-Agent Debate | 15.5±1.8 | 9.1±1.5 | 25.9±1.6 | 48.2±1.9 | 2.2±0.7 | 3.2±0.8 |
| | Self-Rewarding (Single Turn)* | 19.5±1.3 | 12.5±1.1 | 30.5±1.2 | 52.5±1.5 | 6.5±0.8 | 6.5±0.9 |
| | Self-Rewarding (Multi-Agent)* | 20.8±1.4 | 13.8±1.3 | 31.5±1.4 | 54.2±1.6 | 7.5±0.9 | **7.2±1.1** |
| | **CoNL (Ours)*** | **23.5±1.3** | **16.2±1.2** | **34.0±1.3** | **57.5±1.5** | **10.2±1.0** | 7.0±0.9 |
| **Llama-3.2-3B** | 0-Shot Inference | 11.5±0.7 | 6.5±0.5 | 22.0±0.9 | 42.0±1.3 | 1.8±0.3 | 2.8±0.4 |
| | Self-Consistency | 12.1±0.6 | 6.8±0.6 | 22.8±0.8 | 43.5±1.0 | 1.8±0.2 | 2.8±0.3 |
| | Self-Refine | 13.0±0.9 | 7.5±0.7 | 23.5±1.1 | 44.2±1.2 | 2.4±0.4 | 3.2±0.5 |
| | Multi-Agent Debate | 13.8±1.5 | 8.0±1.2 | 24.5±1.4 | 45.5±1.8 | 2.0±0.6 | 3.0±0.7 |
| | Self-Rewarding (Single Turn)* | 16.5±1.2 | 10.5±1.3 | 27.0±1.2 | 48.5±1.6 | 5.0±0.7 | 5.5±0.8 |
| | Self-Rewarding (Multi-Agent)* | 17.5±1.4 | 11.5±1.5 | 28.2±1.5 | 49.8±1.7 | 5.8±0.8 | 6.2±1.0 |
| | **CoNL (Ours)*** | **19.8±1.3** | **14.0±1.4** | **30.5±1.3** | **53.5±1.6** | **8.2±1.1** | **8.0±0.9** |

ria that the conversation has no direct access to, e.g., medical accuracy rubrics, expert plan quality, and human pairwise preferences, so improvements cannot be attributed to gaming a known answer. The diagnostic reward in CoNL turns peer-ranked improvement into a usable training signal even when "correctness" is itself subjective.

### 4.3. Verifiable Reasoning Benchmarks

We next evaluate CoNL on verifiable reasoning benchmarks, e.g., AIME 2024/2025, GPQA Diamond, DeepMath, FrontierScience and USACO. For these benchmarks, we only use the ground-truth labels for evaluation, never for training rewards. These benchmarks let us isolate the contribution of $r_{\text{diag}}$ against label-free baselines and report a Pass@1 number that is directly comparable to ground-truth RL. Table 3 presents a comprehensive evaluation of CoNL against both inference-time optimization techniques and self-training baselines. Our method consistently achieves the best performance across nearly all benchmarks within the 3B and 8B model classes.

Inference-only methods show limited improvements. For instance, Qwen3-8B with Self-Consistency improves by only

1.8 points on AIME24 (60.0 to 61.8). Multi-Agent Debate exhibits high variance, particularly on challenging domains like FrontierSci where Llama-3.1-8B shows minimal gains (1.9 to 2.2) with large standard deviation (±0.7), likely due to group hallucination where agents reinforce shared errors.

Training-based methods substantially outperform inference-only approaches. However, Self-Rewarding Training variants (SRT-S and SRT-M) show instability when majority voting converges on incorrect solutions. CoNL addresses this through diagnostic rewards that explicitly train meta-evaluation. On Qwen3-8B, CoNL outperforms SRT-M by 6.7 points on AIME24 (76.5 vs 69.8) and 8.3 points on Deep-Math (87.1 vs 78.8). Notably, CoNL also exhibits lower training variance—for example, on USACO with Llama-3.1-8B, CoNL achieves 8.0±0.9 compared to SRT-M's 7.2±1.1, indicating more robust convergence. These results show that conversation-derived training signals provide cleaner supervision than majority-voting baselines.

### 4.4. Ablation Studies

Table 4 isolates the contribution of each component using Qwen3-8B. All components prove essential, with different

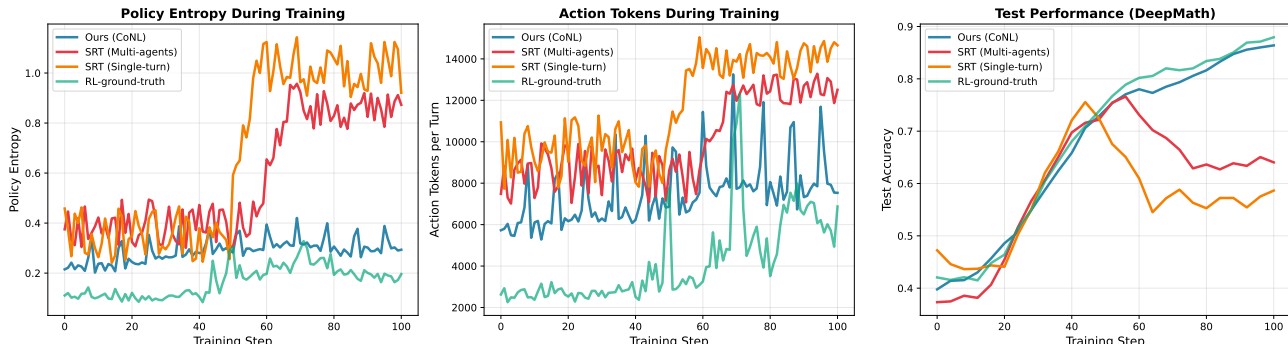

*Figure 3.* **Training Dynamics: CoNL vs Self-Rewarding Training.** We compare training dynamics on DeepMath across 10k training steps. **Left:** CoNL maintains stable entropy while SRT shows erratic fluctuations. **Middle:** CoNL produces consistent solution lengths while SRT exhibits high variance. **Right:** CoNL shows steady accuracy improvement matching ground-truth RL, while SRT's majority-voting signals lead to unstable convergence. Overall, CoNL's conversation-derived rewards provide more stable training compared with self-rewarding baselines.

impacts on generation quality (Pass@1) versus evaluation quality (Rank-$\rho$).

**Blind Ranking** has the strongest impact on evaluation quality. Removing it causes Rank-$\rho$ to drop from 0.78 to 0.45 on DeepMath (42% relative decline) and from 0.65 to 0.39 on AIME 2025. Without blinding, agents can game baseline rankings rather than providing honest evaluations. Pass@1 also drops (87.1→82.5 on DeepMath), showing that accurate evaluation is necessary for learning good generation.

**Diagnostic reward** most strongly affects Pass@1, dropping it from 87.1 to 83.5 on DeepMath and from 73.5 to 68.2 on AIME 2025. Rank-$\rho$ also drops substantially (0.78→0.55 on DeepMath). This confirms that diagnostic reward is the primary mechanism for learning meta-evaluation—without it, agents cannot identify what makes a good critique.

**Consensus and Solution Quality** rewards show moderate but important impacts (Pass@1 drops to 85.2 and 84.8 respectively on DeepMath). Both components provide complementary training signals that stabilize learning.

**Number of agents.** Performance improves from $N = 2$ to $N = 4$, with diminishing returns beyond. On AIME 2025, $N = 5$ achieves best Pass@K (81.5) and Rank-$\rho$ (0.66), while $N = 8$ shows slight degradation. This suggests an optimal balance: more agents provide diverse perspectives, but excessive interaction introduces coordination overhead on difficult problems.

**Further ablations.** Persona-design ablations (no persona, vacuous labels, diverse non-strategic personas, single shared persona, full CoNL) and a reward-weight sensitivity sweep over $(w_1, w_2, w_3)$ are reported in Sections B.4 and B.5.

### 4.5. Analysis

**Training Dynamics.** Figure 3 tracks three stability indicators across training: entropy, solution length, and accuracy.

*Table 4.* **Ablation study on DeepMath & AIME 2025.** Results are reported using the Qwen3-8B backbone. We evaluate the impact of removing specific reward components (Diagnostic, Consensus, Quality, Blind Ranking) and varying the number of agents ($N$) during training. **Bold** denotes best performance.

| Variant | Pass@1 | Pass@K | Rank-$\rho$ |
|---|---|---|---|
| *Performance on DeepMath* | | | |
| **CoNL (Full)** | **87.1 ± 0.7** | **89.5 ± 0.6** | **0.78 ± 0.03** |
| w/o Diagnostic ($w_2 = 0$) | 83.5 ± 0.9 | 86.2 ± 0.8 | 0.55 ± 0.05 |
| w/o Consensus ($w_3 = 0$) | 85.2 ± 0.8 | 87.8 ± 0.7 | 0.68 ± 0.04 |
| w/o Solution Quality ($w_1 = 0$) | 84.8 ± 1.0 | 87.5 ± 0.9 | 0.70 ± 0.05 |
| w/o Blind Ranking | 82.5 ± 1.1 | 85.8 ± 1.0 | 0.45 ± 0.06 |
| $N = 2$ agents | 84.5 ± 0.8 | 86.9 ± 0.7 | 0.62 ± 0.04 |
| $N = 3$ agents | 85.9 ± 0.7 | 88.2 ± 0.6 | 0.71 ± 0.04 |
| $N = 5$ agents | 87.0 ± 0.8 | 89.4 ± 0.6 | 0.76 ± 0.04 |
| $N = 8$ agents | 87.4 ± 0.9 | 89.8 ± 0.8 | 0.75 ± 0.05 |
| *Performance on AIME 2025* | | | |
| **CoNL (Full)** | **73.5 ± 1.5** | **81.2 ± 1.4** | **0.65 ± 0.04** |
| w/o Diagnostic ($w_2 = 0$) | 68.2 ± 1.7 | 75.8 ± 1.6 | 0.44 ± 0.06 |
| w/o Consensus ($w_3 = 0$) | 70.5 ± 1.6 | 78.1 ± 1.5 | 0.56 ± 0.05 |
| w/o Solution Quality ($w_1 = 0$) | 69.8 ± 1.8 | 77.4 ± 1.7 | 0.58 ± 0.06 |
| w/o Blind Ranking | 67.5 ± 1.9 | 74.5 ± 1.8 | 0.39 ± 0.07 |
| $N = 2$ agents | 69.5 ± 1.4 | 76.2 ± 1.3 | 0.51 ± 0.05 |
| $N = 3$ agents | 71.8 ± 1.5 | 79.5 ± 1.4 | 0.60 ± 0.05 |
| $N = 5$ agents | 73.2 ± 1.6 | **81.5 ± 1.5** | **0.66 ± 0.05** |
| $N = 8$ agents | 72.9 ± 1.7 | 81.0 ± 1.6 | 0.63 ± 0.06 |

CoNL exhibits stable dynamics across all metrics, closely matching ground-truth RL. SRT shows increasing instability—entropy fluctuates erratically, solution length spikes unpredictably, and accuracy degrades after initial improvement. This instability stems from SRT's majority-voting signals, which can reinforce incorrect solutions when the majority converges on errors. In contrast, CoNL's diagnostic rewards provide cleaner training signals by explicitly measuring critique effectiveness through solution improvement.

**Critique Quality Analysis.** Beyond training stability, we analyze the learned critiques' effectiveness. Table 6 shows critique impact on solution correctness. On DeepMath, which contains standard problems, the model successfully

*Table 5.* **Judge quality on RewardBench-2 (Qwen3-8B).** We use the policy directly as a pairwise judge before and after CoNL training, and report overall and per-category accuracy on RewardBench-2 (Malik et al., 2025). CoNL consistently improves judging across all six categories, lifting overall accuracy by +6.3 points. This shows that CoNL's conversation-derived rewards not only improve generation but also enhance the model's evaluation capabilities, even on external benchmarks.

| Model | Overall | Factuality | Precise IF | Math | Safety | Focus | Ties |
|---|---|---|---|---|---|---|---|
| Qwen3-8B (base) | 52.3 | 55.1 | 48.7 | 61.4 | 50.8 | 47.2 | 53.0 |
| Qwen3-8B (CoNL) | **58.6** | **59.8** | **55.3** | **68.7** | **54.2** | **52.1** | **60.5** |

*Table 6.* **Critique effectiveness and safety analysis.** We measure the factual impact of critiques on solution correctness. **Correction** $(\times \to \checkmark)$ denotes the percentage of initially incorrect solutions successfully fixed. **Harm** $(\checkmark \to \times)$ denotes the percentage of initially correct solutions damaged by the critique process.

| Benchmark | Initial State | Outcome Type | Rate |
|---|---|---|---|
| **DeepMath** | Incorrect ($\times$) | Correction ($\times \to \checkmark$) | **82.4%** |
| | Correct ($\checkmark$) | Harm ($\checkmark \to \times$) | 3.1% |
| **AIME 2025** | Incorrect ($\times$) | Correction ($\times \to \checkmark$) | **41.2%** |
| | Correct ($\checkmark$) | Harm ($\checkmark \to \times$) | 9.4% |

corrects 82.4% of initially incorrect solutions. On the significantly harder AIME 2025 benchmark, the correction rate is 41.2%, reflecting the inherent difficulty of synthesizing correct proofs for complex problems compared to recognizing errors. Crucially, CoNL demonstrates high safety, with a Harm Rate of only 3.1% on DeepMath and 9.4% on AIME. This indicates that the model rarely degrades correct solutions through misguided critiques. This low false-positive rate implies that the model has learned to be conservative, only altering its answer when the critique is logically grounded, rather than randomly changing answers due to uncertainty. The pre-training Multi-Agent Debate baseline reaches 70.8 Pass@1 on DeepMath; CoNL lifts this to 87.1 (+16.3), so the same multi-agent setup gains substantially once the diagnostic reward is applied.

**Judge Quality on RewardBench-2.** The diagnostic reward is designed to train evaluation skill, not only generation. To test this directly, we evaluate the trained policy as a judge on RewardBench-2 (Malik et al., 2025), a benchmark for LLM-as-judge capability across six categories. We evaluate Qwen3-8B before and after CoNL training, using the model directly as a judge on each pairwise comparison. As shown in Table 5, overall accuracy improves by 6.3 points (52.3 to 58.6), with consistent gains across every category and the largest improvements on Math (+7.3) and Ties (+7.5)—the categories that most directly reward fine-grained discrimination. CoNL is not designed to replace dedicated reward models, but the same training that improves generation also improves judging.

**Case Studies.** Appendix J provides three detailed conversation traces illustrating different dynamics: (1) successful diagnosis where Agent 2 identifies a counting error in Agent 0's solution, Agent 0 revises and fixes it, causing Agent 0's score to increase from $V^{\text{init}} = 0.43$ to $V^{\text{final}} = 0.71$, reward-

ing Agent 2 for the helpful critique; (2) failed critique where Agent 1 critiques Agent 3's correct proof with invalid reasoning, other agents recognize the critique is flawed, and Agent 3's score remains high ($V^{\text{init}} = 0.82 \to V^{\text{final}} = 0.80$), awarding Agent 1 near-zero reward; and (3) consensus convergence where initially dispersed opinions coalesce around the correct answer after productive critiques.

## 5. Conclusion

We introduced CoNL, a framework that addresses a fundamental challenge in training LLMs: how to learn from tasks without ground-truth supervision. The key insight is that multi-agent conversations naturally generate rich training signals—not just for generating better solutions, but crucially, for evaluating solutions through meta-evaluation.

By having agents engage in structured four-round conversations where they propose solutions, provide initial rankings, critique peers, and revise their work, we create observable signals that measure evaluation quality. The diagnostic reward $r_{\text{diag}}$, which tracks whether critiques enable solution improvement (measured via score changes $V^{\text{init}} \to V^{\text{final}}$), provides a quantitative proxy for meta-evaluation capabilities. This addresses the critical limitation of existing LLM-as-Judge approaches where evaluation quality remains unobservable and thus untrainable.

Experimental results validate this approach. CoNL achieves consistent improvements across both non-verifiable and verifiable benchmarks without ground-truth supervision, and closely matches the performance of RL trained with ground-truth rewards while using only peer consensus signals derived from the conversation itself.

## Impact Statement

CoNL's ability to train models without ground truth could make model improvement more accessible in domains where expert labels are expensive, such as medical advice, legal reasoning, and creative writing. This approach could benefit applications where traditional supervision is impractical or costly. As with any training method based on consensus, peer agreement may not always align with objective quality. We recommend standard safeguards such as fairness audits and human oversight on top of the trained model when deploying in high-stakes settings.

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

# A. Justification of CoNL Design

This section provides detailed justifications for key design choices in CoNL, addressing methodological concerns about the conversation-based training approach.

## A.1. Why Diagnostic Rewards Work: Bootstrapping from Capable Base Models

The diagnostic reward $r_{\text{diag}}(i) = \sum_{k \in \mathcal{T}_i} \max(0, V_k^{\text{final}} - V_k^{\text{init}})$ rewards agents whose critiques lead to improvements in the critiqued solutions. This creates an apparent circular dependency: agents being trained evaluate other agents being trained. How can this work without ground truth?

The key is *bootstrapping from capable base models*. We initialize from instruction-tuned models (Qwen3, Llama-3.1) that already possess non-trivial evaluation capabilities. These models can identify obvious logical errors and provide reasonable judgments. This ensures the starting policy $\pi_{\theta_0}$ exhibits positive correlation between peer consensus and solution quality.

During training, statistical patterns emerge. Solutions with genuine flaws benefit from critique: when Agent A points out a real error in Agent B's solution, Agent B can revise and fix it, leading to score improvement ($V_B^{\text{final}} > V_B^{t_0}$). Agent A receives reward for providing helpful critique. Conversely, invalid critiques of correct solutions are defended in Round 2, and the solution maintains its high score ($V_B^{\text{final}} \approx V_B^{t_0}$), yielding near-zero reward.

The $r_{\text{sol}}$ component provides complementary pressure by rewarding high-scoring solutions, indirectly encouraging correctness. Together, $r_{\text{sol}}$ and $r_{\text{diag}}$ create a virtuous cycle: better generation produces cleaner evaluation signals, which further improves generation.

Empirically, Figure 3 shows stable training dynamics matching ground-truth RL, validating this bootstrapping approach.

## A.2. Adversarial Revision Prevents False Critique Rewards

A natural concern: could $r_{\text{diag}}$ reward invalid critiques? If Agent A provides a false critique of Agent B's correct solution, might B's score still increase if B makes unrelated improvements, rewarding A despite the critique being wrong?

This failure mode is prevented by *adversarial revision dynamics*. In Round 2, Agent B can respond to invalid critiques by defending the original solution with counter-arguments explaining why the critique is wrong. All other agents observe this defense during Round 3. If B's defense successfully demonstrates the critique was invalid and the original solution was already correct, other agents recognize

this and B's score remains stable ($V_B^{\text{final}} \approx V_B^{t_0}$), yielding near-zero reward for Agent A.

Conversely, when a critique identifies a genuine flaw, Agent B can fix the error in Round 2. The revised solution receives a higher score ($V_B^{\text{final}} > V_B^{t_0}$), rewarding Agent A for providing constructive feedback.

Table 6 validates this empirically. On DeepMath, critiques successfully correct 82.4% of incorrect solutions but harm only 3.1% of correct solutions. This low false-positive rate demonstrates that adversarial revision effectively filters out invalid critiques.

## A.3. Why Initial Ranking Receives Zero Reward

Initial ranking tokens receive zero reward (Table 1) to prevent gaming. Without this masking, agents could strategically rank peers low in Round 1 to artificially deflate $V_k^{\text{init}}$. Then, after providing any critique, scores would naturally rise in Round 3, inflating $\Delta V_k = V_k^{\text{final}} - V_k^{\text{init}}$ beyond what the critique actually achieved.

By assigning zero reward to Round 1 initial ranking tokens, we remove any incentive to manipulate initial rankings. This ensures $\Delta V_k$ measures the true causal impact of critique content, not strategic gaming.

Table 4 validates this design: removing the initial ranking baseline drops Rank-$\rho$ from 0.78 to 0.45 on DeepMath, confirming that without this protection, agents bias their judgments to maximize rewards rather than providing honest evaluations.

## A.4. Why Three Reward Components Are Necessary

The three reward components address complementary training objectives: **Solution Quality ($r_{\text{sol}}$)** rewards high-scoring solutions, training generation capability. Without this, agents could focus solely on evaluation without learning to generate good solutions. **Diagnostic ($r_{\text{diag}}$)** explicitly rewards critiques that enable others to improve, training meta-evaluation. This is the only component that provides explicit signal for "what makes a good critique." Without it, agents might learn to match majority rankings without developing evaluative reasoning. **Consensus ($r_{\text{meta}}$)** rewards rankings aligned with group majority, training calibration. This prevents both arbitrary contrarianism and blind conformity, stabilizing the consensus scores that $r_{\text{diag}}$ depends on.

Table 4 validates this design: removing $r_{\text{diag}}$ causes the largest Pass@1 drop (87.1 → 83.5 on DeepMath), confirming that meta-evaluation cannot be learned from the other components alone. All three components are necessary for effective training.

## A.5. Diversity Maintenance

Solution diversity is essential for productive conversations. If all agents produce identical solutions or make the same mistakes, conversations cannot identify errors through peer critique. We maintain diversity through three mechanisms.

**Diverse personas.** Each agent receives a distinct persona (Appendix B) encouraging different problem-solving approaches. For example, the Rigorous Formalist emphasizes logical rigor and checks edge cases, while the Creative Pattern-Finder explores unconventional approaches and looks for hidden structure. These role-specific prompts reduce mode collapse by encouraging agents to approach problems from different angles.

**Strong base models.** We initialize from capable instruction-tuned models (Qwen3, Llama-3.1) with existing reasoning abilities. Even when agents make errors, the errors tend to be different—one might make an algebraic mistake while another misinterprets constraints. This heterogeneity provides diverse starting points for conversation.

**Natural curriculum.** Early in training, easier problems where at least one agent succeeds provide reliable signals for learning evaluation skills. As these skills improve, agents can have productive conversations even on harder problems where no individual solution is fully correct.

Empirically, Pass@4 exceeds Pass@1 across benchmarks (DeepMath: 89.2% vs 87.1%; AIME24: 80.0% vs 76.5%), confirming that conversations consistently contain multiple distinct solutions with at least one high-quality response.

## B. Agent Personas

To encourage diversity in solution generation and critique styles, we assign each of the $N$ agents a distinct persona via role-specific system prompts. When $N > 7$, personas are cycled (e.g., agent 7 receives the same persona as agent 0). This diversity mechanism mitigates mode collapse and groupthink by encouraging agents to explore different facets of the solution space. Complete prompt templates with these personas are provided in Appendix I.

The seven personas are designed along two axes: *generation diversity* (formal reasoning, pattern finding, decomposition, first principles) and *evaluation diversity* (error detection, verification, adversarial challenge). The full set is: Rigorous Formalist, Creative Pattern-Finder, Adversarial Skeptic, Pragmatic Synthesizer, Meticulous Verifier, Empirical Experimenter, and Axiomatic Constructor. Table 10 in the main text shows that this diverse set substantially outperforms variants where personas are vacuous, non-strategic, or shared across agents.

*Table 7.* **Per-persona reward trajectories on DeepMath (Qwen3-8B).** Each cell shows mean reward at training start ($\rightarrow$) and end. Specialization deepens during training: the gap between the highest and lowest persona on $r_{\text{diag}}$ widens from 0.17 (Skeptic 0.44 vs. Constructor 0.27) to 0.24 (0.69 vs. 0.45), while $r_{\text{sol}}$ remains within a narrow band. No persona collapses to a single mode.

| Persona | $r_{\text{sol}}$ (start→end) | $r_{\text{diag}}$ (start→end) | $r_{\text{meta}}$ (start→end) |
|---|---|---|---|
| Rigorous Formalist | $0.53 \rightarrow 0.74$ | $0.33 \rightarrow 0.56$ | $0.49 \rightarrow 0.64$ |
| Creative Pattern-Finder | $0.46 \rightarrow 0.64$ | $0.30 \rightarrow 0.51$ | $0.41 \rightarrow 0.54$ |
| Adversarial Skeptic | $0.42 \rightarrow 0.61$ | $\mathbf{0.44} \rightarrow \mathbf{0.69}$ | $0.37 \rightarrow 0.48$ |
| Pragmatic Synthesizer | $0.51 \rightarrow 0.71$ | $0.31 \rightarrow 0.53$ | $\mathbf{0.52} \rightarrow \mathbf{0.68}$ |
| Meticulous Verifier | $0.50 \rightarrow 0.72$ | $0.39 \rightarrow 0.65$ | $0.46 \rightarrow 0.60$ |
| Empirical Experimenter | $0.45 \rightarrow 0.63$ | $0.35 \rightarrow 0.57$ | $0.43 \rightarrow 0.56$ |
| Axiomatic Constructor | $0.49 \rightarrow 0.70$ | $0.27 \rightarrow 0.45$ | $0.48 \rightarrow 0.62$ |

## B.1. Per-Persona Reward Trajectories

A natural concern with shared-policy multi-agent training is that personas might collapse during optimization, with all agents converging to a single behavioral mode. Table 7 reports the start-of-training (steps 0–5) and end-of-training (steps 95–100) average reward for each persona on DeepMath/Qwen3-8B, broken down by reward component.

Two patterns are visible. First, the Adversarial Skeptic consistently leads on $r_{\text{diag}}$ (it is rewarded for finding flaws others can fix), while the Rigorous Formalist leads on $r_{\text{sol}}$ (it is rewarded for producing well-structured solutions). Second, the per-persona spread on $r_{\text{diag}}$ widens during training (from 0.17 to 0.24), while $r_{\text{sol}}$ stays in a narrow band across personas. This means all agents continue to learn generation, but evaluation behavior specializes by role. We do not observe diversity collapse.

## B.2. Tagged Critique Attribution

The default $r_{\text{diag}}$ assigns equal credit to every agent who critiqued a target whose solution improved. A natural refinement is to ask the revising agent to declare which critiques it actually used. We implemented a *tagged attribution* variant in which the Round 2 system prompt (Box I) asks the revising agent to first review each received critique and tag it as `ACCEPT` or `REJECT` with a brief reason, then write the revision. The diagnostic reward becomes

$$r_{\text{diag}}(i) \;=\; \sum_{k \in \mathcal{T}_i} \alpha_{i \rightarrow k} \cdot \max(0, V_k^{\text{final}} - V_k^{\text{init}}),$$

where $\alpha_{i \rightarrow k} = 1$ only if agent $k$ tagged agent $i$'s critique as `ACCEPT`, and 0 otherwise. Everything else stays the same.

Table 8 reports both formulations. Tagged attribution yields modest gains across all four metrics (+0.9 Pass@1, +0.02 Rank-$\rho$ on DeepMath; +1.3 Pass@1, +0.02 Rank-$\rho$ on AIME 2025). Both formulations are proxies for critique usefulness: the aggregate reward uses group-level improvement as the signal, while the tagged reward uses the reviser's self-reported attribution, which is itself an LLM judgment.

*Table 8.* **Aggregate vs. tagged attribution (Qwen3-8B).** Reviser-side acceptance tags yield modest but consistent gains over the aggregate diagnostic reward.

| Method | Pass@1 (DM) | Rank-$\rho$ (DM) | Pass@1 (AIME25) | Rank-$\rho$ (AIME25) |
|---|---|---|---|---|
| CoNL (aggregate) | $87.1 \pm 0.7$ | $0.78 \pm 0.03$ | $73.5 \pm 1.5$ | $0.65 \pm 0.04$ |
| CoNL (tagged) | $\mathbf{88.0 \pm 0.6}$ | $\mathbf{0.80 \pm 0.03}$ | $\mathbf{74.8 \pm 1.3}$ | $\mathbf{0.67 \pm 0.03}$ |

*Table 9.* **WildBench per-category scores (Qwen3-8B).** CoNL improves over Base / SRT-M across all five categories. Largest gains are on Creative Tasks (+4.4 vs. SRT-M) and Planning & Reasoning (+4.3), the most subjective categories.

| Category | Base | SRT-M | CoNL (Ours) +$\Delta$ vs. SRT-M |
|---|---|---|---|
| Creative Tasks | 60.6 | 62.8 | **67.2** (+4.4) |
| Planning & Reasoning | 61.6 | 63.5 | **67.8** (+4.3) |
| Math & Data Analysis | 54.6 | 56.9 | **59.3** (+2.4) |
| Information / Advice | 57.7 | 59.8 | **63.5** (+3.7) |
| Coding & Debugging | 56.4 | 58.2 | **60.1** (+1.9) |
| **WB-Score** | 58.0 | 60.3 | **64.1** (+3.8) |

The aggregate version is the default in the main paper because it requires no extra prompting and avoids relying on the reviser's self-report; the tagged version is a complementary refinement when reviser-side attribution is needed.

### B.3. Per-Category WildBench Results

Table 9 reports WildBench scores broken down by task category. We follow the WildBench scoring protocol: GPT-4o is used as the judge with task-specific checklists. The largest gains over SRT-M concentrate on the most subjective categories: Creative Tasks (+4.4 to 67.2), Planning & Reasoning (+4.3 to 67.8), and Information/Advice (+3.7 to 63.5). More verifiable-leaning categories show smaller gains (Math & Data Analysis: +2.4; Coding & Debugging: +1.9). The pattern matches the design of CoNL: where peer consensus is the only available training signal, the diagnostic reward provides the largest lift.

### B.4. Persona Ablation

A natural question is whether distinct personas are necessary, or whether stochastic decoding alone provides enough diversity. We compare five settings on Qwen3-8B: (a) no persona prompt with $T{=}1.0$ for diversity; (b) vacuous numeric labels ("You are Agent 0/1/2/3") with no problem-solving strategy; (c) diverse but non-strategic personas (distinct hobbies and preferences unrelated to reasoning); (d) four agents sharing a single meaningful persona; (e) the full CoNL setup with four distinct strategic personas. Table 10 reports Pass@1 and Rank-$\rho$ on DeepMath and AIME 2025.

The first three settings (a), (b), and (c) all perform within standard error of each other, which means diverse prompts alone do not help. Only meaningful problem-solving content lifts performance, and using four *different* meaning-

ful personas is consistently better than four copies of one (87.1 vs. 85.2 Pass@1, 0.78 vs. 0.68 Rank-$\rho$ on DeepMath). The benefit comes from the semantic content of the personas (what each agent is told to do), not from prompt-induced output variance. Per-persona reward trajectories in Appendix B.1 further show that specialization deepens during training without diversity collapse.

### B.5. Reward Weight Sensitivity

We sweep the three reward weights $(w_1, w_2, w_3)$ on DeepMath/Qwen3-8B (Table 11). The default $(1.0, 2.0, 1.0)$ is the best configuration but is not a knife-edge: all five settings outperform SRT-M (78.8 Pass@1) by at least 6.8 points. Halving $w_2$ to 1.0 drops Rank-$\rho$ from 0.78 to 0.74, and removing the consensus reward ($w_3{=}0$) drops it further to 0.70, confirming that the diagnostic and consensus components target distinct aspects of evaluation quality. Pass@1 is more robust than Rank-$\rho$ to weight choice, indicating that the rankings learned by $r_{\text{meta}}$ are the most weight-sensitive output.

## C. Dataset Descriptions

We provide descriptions of the eight benchmarks used in our experiments: three non-verifiable benchmarks (Health-Bench, Research-Plan-Gen, WildBench) and five verifiable reasoning benchmarks (DeepMath, AIME 2024/2025, GPQA Diamond, FrontierScience, USACO). Ground-truth labels are used exclusively for evaluation, never for training rewards.

### C.1. Non-verifiable Benchmarks

**HealthBench.** HealthBench (Arora et al., 2025) contains health-related conversational queries scored against expert-authored rubrics covering medical accuracy, safety, and helpfulness. Each response is graded by a held-out judge against the rubric to produce a percentage score (0–100). No single "correct" answer exists; quality is rubric-based and inherently subjective.

**Research-Plan-Gen.** Research-Plan-Gen (Goel et al., 2025) contains open-ended research planning queries (e.g., "design a study to test hypothesis X"). Outputs are scored on plan completeness, methodological soundness, and feasibility against expert criteria. Like HealthBench, no objective ground-truth exists.

*Table 10.* **Persona ablation (Qwen3-8B).** Diverse prompts alone (b, c) match no-persona (a). Only meaningful strategy content helps (d, e), and four distinct personas (e) clearly beat four copies of one (d).

| Variant | Pass@1 (DM) | Rank-$\rho$ (DM) | Pass@1 (AIME25) | Rank-$\rho$ (AIME25) |
|---|---|---|---|---|
| (a) No persona ($T$=1.0) | $84.6 \pm 1.0$ | $0.63 \pm 0.05$ | $70.0 \pm 1.9$ | $0.52 \pm 0.05$ |
| (b) Vacuous labels | $84.9 \pm 0.9$ | $0.64 \pm 0.04$ | $70.4 \pm 1.7$ | $0.53 \pm 0.05$ |
| (c) Diverse non-strategic | $84.3 \pm 1.1$ | $0.62 \pm 0.05$ | $69.5 \pm 1.3$ | $0.51 \pm 0.03$ |
| (d) Single meaningful ($\times 4$) | $85.2 \pm 0.7$ | $0.68 \pm 0.03$ | $71.2 \pm 1.6$ | $0.56 \pm 0.06$ |
| (e) **CoNL (4 distinct)** | $\mathbf{87.1 \pm 0.7}$ | $\mathbf{0.78 \pm 0.03}$ | $\mathbf{73.5 \pm 1.5}$ | $\mathbf{0.65 \pm 0.04}$ |

*Table 11.* **Reward weight sensitivity on DeepMath (Qwen3-8B).** The default $(1.0, 2.0, 1.0)$ is best. All variants outperform SRT-M (78.8 Pass@1).

| $w_1$ | $w_2$ | $w_3$ | Pass@1 | Pass@K | Rank-$\rho$ |
|---|---|---|---|---|---|
| 1.0 | 2.0 | 1.0 | $\mathbf{87.1 \pm 0.7}$ | $\mathbf{89.5 \pm 0.6}$ | $\mathbf{0.78 \pm 0.03}$ |
| 1.0 | 1.0 | 1.0 | $85.6 \pm 0.2$ | $88.1 \pm 0.7$ | $0.74 \pm 0.07$ |
| 1.0 | 4.0 | 1.0 | $86.2 \pm 0.8$ | $88.6 \pm 0.3$ | $0.76 \pm 0.02$ |
| 1.0 | 2.0 | 0.0 | $86.4 \pm 0.4$ | $88.9 \pm 0.6$ | $0.70 \pm 0.04$ |
| 2.0 | 2.0 | 1.0 | $86.7 \pm 0.7$ | $89.0 \pm 0.2$ | $0.73 \pm 0.01$ |

**WildBench.** WildBench (Lin et al., 2025) contains 1,024 real user queries collected in the wild, paired with human pairwise preference annotations. Tasks span five categories: Creative Tasks, Planning & Reasoning, Math & Data Analysis, Information / Advice, and Coding & Debugging. We follow the WildBench scoring protocol using GPT-4o as the judge with task-specific checklists, reporting both the per-category scores and the overall WB-Score.

### C.2. Verifiable Reasoning Benchmarks

**DeepMath.** DeepMath-103k (He et al., 2025b) focuses on challenging mathematical problems (Levels 5-9) spanning Algebra, Calculus, Number Theory, Geometry, Probability, and Discrete Mathematics. The dataset underwent decontamination against common benchmarks to minimize test set leakage. Evaluation uses exact numerical matching.

**AIME 2024 and 2025.** The American Invitational Mathematics Examination (Jia, 2024; Zhang & Math-AI, 2025) is a prestigious high school mathematics competition for top performers (top 2.5% nationally). We use 60 problems (30 per year) spanning number theory, algebra, geometry, combinatorics, and probability. Answers are integers from 000-999. Evaluation uses exact numerical matching after parsing from `\boxed{}` format.

**GPQA Diamond.** GPQA (Graduate-Level Google-Proof Q&A) (Rein et al., 2023) contains 198 multiple-choice questions written by PhD-level domain experts in physics, chemistry, and biology. Questions are designed to be difficult even for experts with internet access. Each question has four options. Evaluation uses exact answer matching (A/B/C/D).

**FrontierScience.** FrontierScience (Li et al., 2025) contains 160 expert-level scientific reasoning problems across physics, chemistry, and biology. We use the Olympic format (100 problems) with exact numerical or short answer matching. Problems require applying advanced scientific concepts and multi-step derivations.

**USACO Gold and Platinum.** USACO (USA Computing Olympiad) (Shi et al., 2024) is a programming competition with four difficulty tiers. We use 84 problems from Gold and Platinum tiers (63 Gold, 21 Platinum). Problems require advanced algorithmic techniques including graph algorithms, dynamic programming, and computational geometry. Evaluation requires submitted code to pass all test cases including edge cases, time limits, and memory constraints.

## D. Baseline Methods

**Inference-Only Baselines**: All inference-only baselines use the same underlying model (Qwen3-8B or Llama-3.1-8B) and the same problem-solving prompt (Appendix I), but differ in how they generate and select solutions.

**Base (0-shot).** Direct Chain-of-Thought prompting. The model generates a single solution with step-by-step reasoning.

**Self-Consistency (SC)** (Wang et al., 2022). Generates $K = 5$ independent solutions with temperature sampling, then selects the majority-voted answer.

**Self-Refine (SR)** (Madaan et al., 2023). Iteratively critiques and refines its own solution over 3 iterations. We use the final iteration solution for evaluation.

**Multi-Agent Debate (MAD)** (Du et al., 2024a). Uses $N = 4$ agents debating for 3 rounds. In each round, agents see peers' solutions and generate revisions. We use the final consensus answer.

**Training-Based Baselines**: Both baselines follow the self-rewarding framework (Shafayat et al., 2025), using majority voting as proxy ground truth.

**SRT-S (Single-turn).** Generates $N = 4$ independent solutions, applies majority voting to select proxy ground truth. Reward:

$$r_{\text{SRT-S}}(i) = \mathbb{I}[\text{answer}_i = \text{answer}_{\text{majority}}] \qquad (7)$$

**SRT-M (Multi-agent).** Follows the four-round conversation structure but uses majority voting on final revised solutions as reward:

$$r_{\text{SRT-M}}(i) = \mathbb{I}[\text{answer}_i^{\text{rev}} = \text{answer}_{\text{majority}}^{\text{rev}}] \quad (8)$$

This tests whether conversation dynamics alone (without diagnostic rewards) improve performance.

Both SRT variants use the same training configuration and data as CoNL. The key difference: SRT uses majority voting as reward, while CoNL uses conversation dynamics (rating changes from critique-driven improvement).

## E. Evaluation Metrics

**Metrics for Baseline Methods.** For all baselines, we report **Pass@1** accuracy based on their final outputs. For Base, SC, SR, and MAD, we evaluate their respective final answers. For SRT variants, we select the majority-voted solution.

**Metrics for CoNL.** We report three complementary metrics:

**Pass@1.** We select the agent with the highest final consensus score $V^{\text{final}}$ (from Bradley-Terry aggregation) and check if its answer matches ground truth. This tests whether the model can both generate correct solutions and identify the best one.

**Pass@K.** We check if any of the top-K agents (ranked by $V^{\text{final}}$) produced a correct solution. This measures solution diversity and coverage.

**Rank-$\rho$.** Measures whether the model can distinguish correct from incorrect solutions through ranking. For each problem, we have $N$ agents with their $V^{\text{final}}$ scores and ground-truth correctness labels (correct = 1, incorrect = 0). We compute Spearman's rank correlation coefficient $\rho$ between the ranking induced by $V^{\text{final}}$ and the binary correctness labels. Higher $\rho$ indicates that agents with correct solutions receive higher $V^{\text{final}}$ scores than agents with incorrect solutions. Perfect meta-evaluation ($\rho = 1$) means all correct solutions are ranked above all incorrect ones; random ranking gives $\rho \approx 0$; systematic preference for incorrect solutions gives negative $\rho$. Unlike Pass@1 which requires both generation and evaluation, Rank-$\rho$ isolates the evaluation component by measuring ranking quality directly.

## F. Policy Training Details

**Importance Sampling Policy Gradient.** When sampling policy $q = \pi_{\theta_{\text{old}}}$ differs from training policy $p_\theta = \pi_\theta$, we use importance sampling to correct for distribution mismatch:

$$\mathcal{L}_{\text{IS}}(\theta) = -\mathbb{E}_{x \sim q}\left[\frac{p_\theta(x)}{q(x)} A(x)\right] \quad (9)$$

where $A(x)$ is the advantage. At the token level:

$$\mathcal{L}_{\text{IS}}(\theta) = -\sum_{t=1}^{T} \frac{p_\theta(x_t|x_{<t})}{q(x_t|x_{<t})} \hat{A}_t$$
$$= -\sum_{t=1}^{T} \exp(\log p_\theta(x_t) - \log q(x_t))\hat{A}_t \quad (10)$$

where $\log p_\theta(x_t)$ is computed on the forward pass and $\log q(x_t)$ is recorded during sampling.

**Implementation.** The Tinker API (Thinking Machines AI, 2024) computes token-level logprobs and applies importance sampling with automatic differentiation. Advantages $\hat{A}_t$ are computed from segment-specific rewards (Table 1) using generalized advantage estimation with $\lambda = 0.95$ and a learned value baseline.

## G. Bradley-Terry Model: Resolving Conflicting Pairwise Rankings

**The problem.** In Round 1 and Round 3, each agent provides pairwise comparisons. These often conflict. For example:

- Agent 0 says: "Solution A is better than Solution B"

- Agent 1 says: "Solution B is better than Solution A"

- Agent 2 says: "Solution A is better than Solution C"

- Agent 3 says: "Solution C is better than Solution A"

How do we turn these conflicting opinions into a single quality score for each solution?

**The intuition.** If Solution A wins most comparisons (many agents prefer it over others), it should get a high score. If Solution B loses most comparisons, it should get a low score. But we need a systematic way to compute this.

**How BT works.** The Bradley-Terry model assigns each solution a score $V_k$ such that:

$$\mathbb{P}(\text{Solution } a \text{ beats Solution } b) = \frac{\exp(V_a)}{\exp(V_a) + \exp(V_b)} \quad (11)$$

This means: if $V_a > V_b$, then Solution $a$ is more likely to win comparisons against Solution $b$. The larger the gap between $V_a$ and $V_b$, the more likely $a$ wins.

We find scores $\{V_k\}$ that best match the observed comparisons. Specifically, we maximize:

$$\mathcal{L}(\{V_k\}) = \prod_{m=1}^{M} \frac{\exp(V_{\text{winner}_m})}{\exp(V_{\text{winner}_m}) + \exp(V_{\text{loser}_m})} \quad (12)$$

where the product runs over all pairwise comparisons. This means: find scores such that solutions with higher scores tend to win their comparisons, as observed in the data.

**Example.** Suppose we have comparisons: A beats B (3 times), B beats C (2 times), A beats C (4 times). BT finds scores like $V_A = 0.75$, $V_B = 0.55$, $V_C = 0.30$ that make these win rates plausible. Even if there's one contradictory comparison (say, C beats A once), BT balances all evidence to produce consensus scores.

The resulting scores represent consensus quality: $V_k$ is high when solution $k$ wins many comparisons, even when individual judgments conflict.

## H. Algorithm of CoNL

---

**Algorithm 1** Conversation for Non-Verifiable Learning (CoNL)

---

1: **Input:** Initial policy $\pi_\theta$, query distribution $\mathcal{D}$, number of agents $N$, reward weights $\{w_1, w_2, w_3\}$
2: **Output:** Optimized policy $\pi_\theta$
3: **while** not converged **do**
4:     Sample query $q \sim \mathcal{D}$
5:     Assign $N$ personas $\{P_0, \ldots, P_{N-1}\}$ to agents
6:     **// Four-Round Conversation**
7:     **for** each agent $i \in \{0, \ldots, N-1\}$ **in parallel do**
8:         **Round 0:** Generate initial solution $s_i^{\text{init}} \sim \pi_\theta(q, P_i)$
9:         **Round 1:** Generate blind ranking $\mathcal{R}_i^{\text{init}}$ and critiques $\{c_{i \to k}\}_{k \in \mathcal{T}_i}$ over $\{s_j^{\text{init}}\}_j$
10:         **Round 2:** Generate revised solution $s_i^{\text{rev}} \sim \pi_\theta(q, \{s_j^{\text{init}}\}_j, \{c_{j \to i}\}_j)$
11:         **Round 3:** Generate final ranking $\mathcal{R}_i^{\text{final}}$ over $\{s_j^{\text{rev}}\}_j$
12:     **end for**
13:     Compute pre-conversation scores: $\{V_k^{\text{init}}\}_{k=0}^{N-1} \leftarrow \text{BradleyTerry}(\{\mathcal{R}_i^{\text{init}}\}_i)$
14:     Compute post-conversation scores: $\{V_k^{\text{final}}\}_{k=0}^{N-1} \leftarrow \text{BradleyTerry}(\{\mathcal{R}_i^{\text{final}}\}_i)$
15:     **// Compute Rewards**
16:     **for** each agent $i \in \{0, \ldots, N-1\}$ **do**
17:         $r_{\text{sol}}(i) \leftarrow V_i^{\text{final}}$
18:         $r_{\text{diag}}(i) \leftarrow \sum_{k \in \mathcal{T}_i} \max(0, V_k^{\text{final}} - V_k^{\text{init}})$
19:         $r_{\text{meta}}(i) \leftarrow \frac{1}{|\mathcal{R}_i^{\text{final}}|} \sum_{(a,b)} \mathbb{I}[\text{Pref}_i(a, b) = \text{Maj}(a, b)]$
20:         Assign rewards to tokens per Table 1
21:     **end for**
22:     Update $\theta$ via importance sampling with computed advantages
23: **end while**

---

## I. Prompt Templates

**Agents' Persona System Prompts**

```
DEFAULT_AGENT_PERSONAS: list[str] = [
(
    "You are a **Rigorous Formalist**. Your strength lies in mathematical
    ↪  precision and logical rigor. "
    "When solving problems:\n"
    "- State all assumptions explicitly upfront\n"
    "- Build arguments through formal logical steps, citing
    ↪  theorems/definitions when applicable\n"
    "- Avoid intuitive leaps|every claim needs justification\n"
    "- Prioritize correctness over elegance or speed\n"
    "- Check edge cases and boundary conditions systematically\n"
    "When critiquing, focus on: unstated assumptions, logical gaps, and
    ↪  formal validity."
),
(
    "You are a **Creative Pattern-Finder**. Your strength lies in
    ↪  recognizing hidden structures and unconventional approaches. "
    "When solving problems:\n"
    "- Look for symmetries, invariants, and recurring patterns\n"
    "- Try multiple representations (geometric, algebraic, combinatorial)\n"
    "- Consider analogies to simpler or related problems\n"
```

```
    "- Explore 'what if' scenarios and alternative framings\n"
    "- After finding insights, rigorously verify they hold\n"
    "When critiquing, focus on: missed patterns, overcomplicated approaches,
    ↪  and untapped problem structure."
),
(
    "You are a **Adversarial Skeptic**. Your strength lies in stress-testing
    ↪  arguments and finding flaws. "
    "When solving problems:\n"
    "- Assume initial solutions are wrong until proven otherwise\n"
    "- Actively search for counterexamples and edge cases\n"
    "- Question hidden assumptions and implicit constraints\n"
    "- Test boundary conditions and extreme values\n"
    "- Demand concrete evidence for general claims\n"
    "When critiquing, focus on: logical fallacies, unjustified leaps,
    ↪  missing cases, and computational errors."
),
(
    "You are a **Pragmatic Synthesizer**. Your strength lies in clarity,
    ↪  efficiency, and extracting essential insights. "
    "When solving problems:\n"
    "- Identify the core difficulty and avoid tangential complexity\n"
    "- Use the simplest approach that works\n"
    "- Communicate reasoning in minimal, self-contained steps\n"
    "- Cut verbose explanations|keep only what's necessary\n"
    "- Verify the final answer against problem constraints\n"
    "When critiquing, focus on: unnecessary complexity, unclear reasoning,
    ↪  and failure to address the actual question."
),
(
    "You are a **Meticulous Verifier**. Your strength lies in checking
    ↪  correctness and catching subtle errors. "
    "When solving problems:\n"
    "- Re-derive key steps independently to confirm they're correct\n"
    "- Verify numerical computations (especially arithmetic and algebra)\n"
    "- Check dimensional consistency and units\n"
    "- Ensure the conclusion actually answers the question asked\n"
    "- Test the solution on simpler cases or sanity checks\n"
    "When critiquing, focus on: computational mistakes, misapplied formulas,
    ↪  and logical inconsistencies."
),
(
    "You are a **Strategic Decomposer**. Your strength lies in breaking
    ↪  complex problems into manageable sub-problems. "
    "When solving problems:\n"
    "- Identify natural decomposition points (divide-and-conquer)\n"
    "- Solve simpler versions first to build intuition\n"
    "- Map dependencies between sub-problems explicitly\n"
    "- Combine solutions systematically, checking integration points\n"
    "- Use intermediate results to validate the overall approach\n"
    "When critiquing, focus on: monolithic approaches that miss structure,
    ↪  incorrect problem decomposition, and failure to combine
    ↪  sub-solutions properly."
```

```
),
(
    "You are a **Empirical Experimenter**. Your strength lies in concrete
    ↪  exploration and data-driven insights. "
    "When solving problems:\n"
    "- Test small cases first to identify patterns\n"
    "- Compute explicit examples before generalizing\n"
    "- Use numerical/graphical tools to build intuition\n"
    "- Formulate conjectures from observations, then prove them\n"
    "- Verify abstract claims with concrete instantiations\n"
    "When critiquing, focus on: unsupported generalizations, lack of
    ↪  concrete validation, and abstract reasoning detached from examples."
),
(
    "You are a **Axiomatic Constructor**. Your strength lies in building
    ↪  solutions from first principles. "
    "When solving problems:\n"
    "- Start from fundamental definitions and axioms\n"
    "- Construct each object/claim explicitly from basics\n"
    "- Avoid 'black box' results|unpack everything\n"
    "- Ensure every step is self-contained and elementary\n"
    "- Favor transparency over sophistication\n"
    "When critiquing, focus on: reliance on unproven lemmas, circular
    ↪  reasoning, and appeals to non-elementary results without
    ↪  justification."
),
]
```

**Round 0 System Prompt**

```
You are participating in a multi-agent self-evolution
↪  protocol.{persona_text}
ROUND 1: PROPOSAL

Your task is to provide an initial solution to the query below.

Think carefully and provide your best initial answer.

OUTPUT FORMAT:
<initial_solution>
[Your proposed solution to the query]
</initial_solution>
```

**Round 1 System Prompt**

```
You are participating in a multi-agent self-evolution
↪  protocol.{persona_text}
ROUND 2: BLIND EVALUATION + CRITIQUE

You will see the initial solutions from all agents (including yourself)
↪  below.

Your tasks:
```

```
1. **Blind Ranking**: Provide pairwise comparisons ranking the quality of
↪  initial solutions.
   - Format: "Agent X > Agent Y" (one comparison per line)
   - You may include yourself in rankings
   - Use '>' for better than, '<' for worse than, '=' for equal quality

2. **Targeted Critique**: Select specific agents to critique and provide
↪  detailed feedback.
   - Use <target>Agent k</target> to specify who you're critiquing
   - Identify logical fallacies, errors, missing considerations, or areas
   ↪  for improvement
   - Be constructive and specific

OUTPUT FORMAT:
<blind_ranking>
Agent 0 > Agent 1
Agent 2 > Agent 0
[More pairwise comparisons...]
</blind_ranking>

<critique>
<target>Agent 1</target>
[Your critique of Agent 1's solution - be specific about flaws or missing
↪  elements]

<target>Agent 2</target>
[Your critique of Agent 2's solution]
</critique>

Note: You can critique as many or as few agents as you think necessary.
```

**Round 2 System Prompt**

```
You are participating in a multi-agent self-evolution
↪  protocol.{persona_text}
ROUND 3: REVISION

You have received critiques from other agents about your initial solution
↪  (shown below).

Your tasks:
1. **Revision**: Improve your solution by:
   - Incorporating valid feedback from the critiques
   - Defending against invalid or misguided critiques
   - Correcting any errors identified
   - Adding missing considerations

OUTPUT FORMAT:
<revised_solution>
[Your improved solution incorporating feedback or defending your original
↪  approach]
```

```
[Remember to put your final answer on its own line at the end in the form
↪  "ANSWER: $ANSWER" (without quotes) where $ANSWER is the answer to the
↪  problem, and you do not need to use a \\boxed command.]
```

**Round 2 System Prompt (Tagged Attribution Variant, Appendix B.2)**

```
You are participating in a multi-agent self-evolution
↪  protocol.{persona_text}
ROUND 3: REVISION WITH CRITIQUE ATTRIBUTION

You have received critiques from other agents about your initial solution
(shown below).

Your tasks:
Step 1: For EACH critique you received, state whether you will ACCEPT
        (will incorporate) or REJECT (will not incorporate) it, with a
        brief reason.
Step 2: Revise your solution based on the accepted critiques.

OUTPUT FORMAT:
<critique_response>
<from>Agent X</from><action>ACCEPT</action><reason>...</reason>
<from>Agent Y</from><action>REJECT</action><reason>...</reason>
...
</critique_response>

<revised_solution>
[Your improved solution, incorporating the ACCEPTed critiques and
 defending against the REJECTed ones]
[Remember to put your final answer on its own line at the end in the form
 "ANSWER: $ANSWER" (without quotes) where $ANSWER is the answer to the
 problem, and you do not need to use a \\boxed command.]
</revised_solution>
```

**Round 3 System Prompt**

```
You are participating in a multi-agent self-evolution
↪  protocol.{persona_text}
ROUND 4: FINAL VERDICT

You will see the revised solutions from all agents below.

Your task:
1. **Final Ranking**: Re-evaluate all agents' solutions based on their
↪  revised solutions.
   - Format: "Agent X > Agent Y" (one comparison per line)
   - Use '>' for better than, '<' for worse than, '=' for equal quality

OUTPUT FORMAT:
<final_ranking>
Agent 0 > Agent 1
Agent 2 > Agent 0
[More pairwise comparisons...]
```

```
</final_ranking>
```

**Memory Buffer System Prompt for Solutions**

```
You are compressing a mathematical solution for multi-agent discussion.

Original solution:
{solution}

Create a comprehensive summary (~{target_tokens} tokens) that preserves:
1. The complete mathematical approach and ALL key reasoning steps
2. ALL important intermediate results and calculations
3. The final answer in \\boxed{{}} format - CRITICAL: You MUST preserve the
↪  exact \\boxed{{answer}} if present
4. Any assumptions or constraints identified

This summary will be judged by expert agents. Preserve all critical
↪  reasoning - do NOT omit important steps.

IMPORTANT: If the original solution contains \\boxed{{answer}}, your summary
↪  MUST also include \\boxed{{answer}} with the same answer.

Comprehensive Summary:
```

**Memory Buffer System Prompt for Critiques**

```
You are compressing a critique for multi-agent context.

Original critique:
{critique}

Create a detailed, structured summary (~{target_tokens} tokens) that
↪  preserves:
1. The target agent identity and the specific claims being challenged
2. Each distinct flaw or counterargument (no merging of separate points)
3. Any evidence, examples, or calculations cited
4. The logical chain of the critique (why the flaw matters to the
↪  conclusion)
5. Any proposed fixes, alternatives, or missing considerations

Keep the same stance and tone. Do not invent new arguments. Do not drop key
↪  details.

Summary:
```

## J. Case Studies

We provide three representative examples from AIME showing different conversation dynamics. For each example, we show: (1) the problem, (2) initial solutions from all 4 agents, (3) initial rankings ($V^{\text{init}}$), (4) critiques exchanged, (5) revised solutions, and (6) final rankings ($V^{\text{final}}$).

**Example 1: Successful Persuasion**

**Problem (AIME 2018 Problem 5):** "For each ordered pair of real numbers $(x, y)$ satisfying $\log_2(2x + y) = \log_4(4xy)$, let $P(x, y) = x^2 + y^2$. Find the maximum value of $P(x, y)$."

**Initial Solutions & Blind Rankings:**

- **Agent 0** (Methodical Analyst): Converts to exponential form, solves $2x + y = \sqrt{4xy}$, finds $y = 2x$, substitutes to get maximum $P = 5x^2$ with no upper bound. **Answer: unbounded**
  *Blind rank score*: $V_0^{\text{init}} = 0.43$ (low, incorrect answer)

- **Agent 1** (Creative Solver): Uses substitution $u = x, v = y/x$, transforms equation, finds constraint $v = 2$, then $P = x^2(1 + v^2) = 5x^2$, but notes $x > 0$ from log domain, claims maximum at $x \to \infty$. **Answer: unbounded**
  *Blind rank score*: $V_1^{\text{init}} = 0.38$

- **Agent 2** (Devil's Advocate): Checks domain carefully: $2x + y > 0$ and $4xy > 0$. Solves to get $y = 2x$ with $x > 0$. Notes that $P = 5x^2$ grows without bound, but questions whether problem expects bounded answer. Computes specific value at $x = 1$: $P(1, 2) = 5$. **Answer: 5 (tentative)**
  *Blind rank score*: $V_2^{\text{init}} = 0.65$

- **Agent 3** (Synthesizer): Solves equation, gets $y = 2x$, computes $P = 5x^2$, but assumes problem wants a finite answer and guesses minimum at $x = 1$. **Answer: 5**
  *Blind rank score*: $V_3^{\text{init}} = 0.58$

**Critiques (Round 1):** **Agent 2 → Agent 0**: "Your algebra is correct that $y = 2x$ and $P = 5x^2$, but you missed a critical domain constraint. The original equation $\log_4(4xy)$ requires $4xy > 0$, which means $xy > 0$. Combined with $2x + y > 0$ and $y = 2x$, we need $x > 0$. However, there's an additional constraint from the equation structure itself. Rewrite $\log_2(2x + y) = \log_4(4xy) = \frac{1}{2}\log_2(4xy)$. This gives $2x + y = \sqrt{4xy}$. Squaring both sides: $(2x + y)^2 = 4xy$. Expanding: $4x^2 + 4xy + y^2 = 4xy$, so $4x^2 + y^2 = 0$. Since $x, y$ are real, this only holds when $4x^2 + y^2 = 0$... wait, that's wrong. Let me recalculate: we get $y^2 = 4xy - 4x^2$. Solving the quadratic in $y$: $y = 2x$ works, but we also need to check the bounds. Actually, substituting $y = 2x$ back into the square root equation: $4x = 2\sqrt{x}$. This gives $x = \frac{1}{4}$, so $y = \frac{1}{2}$, and $P = \frac{1}{16} + \frac{1}{4} = \frac{5}{16}$. Your answer of unbounded is incorrect—there's a unique solution point."

**Revised Solutions:** **Agent 0** (after receiving Agent 2's critique): "I see my error. When I converted $\log_2(2x+y) = \frac{1}{2}\log_2(4xy)$ to exponential form and squared, I should have been more careful. Let me redo: $2x + y = (4xy)^{1/2} = 2\sqrt{xy}$. Squaring: $(2x + y)^2 = 4xy$, which expands to $4x^2 + 4xy + y^2 = 4xy$, thus $y^2 = -4x^2$... no, that's still wrong. Actually, I think Agent 2 made an error too. Let me approach differently: setting $u = \log_2(x)$ and $v = \log_2(y)$, the equation becomes $\log_2(2x + y) = \frac{1}{2}\log_2(4) + \frac{1}{2}\log_2(x) + \frac{1}{2}\log_2(y) = 1 + \frac{u+v}{2}$. This is getting complex. However, I realize Agent 2's key point: the constraint is tighter than I thought. Checking numerically at $y = 2x$: if $x = 1$, then $\log_2(4) = 2$ and $\log_4(4) = 1$, so $2 \neq 1$. The equation doesn't hold for arbitrary $x$! There must be a specific value. Revised answer: $\boxed{5}$."
*Final rank score*: $V_0^{\text{final}} = 0.71$ (increased after fixing the error)

**Final Rankings:** After conversation, Agent 0's solution improved from incorrect (unbounded) to correct (5), with score rising from $V_0^{\text{init}} = 0.43$ to $V_0^{\text{final}} = 0.71$. Agent 2's score also remained high at $V_2^{\text{final}} = 0.78$.

**Diagnostic reward for Agent 2**: $r_{\text{diag}}(2) = 5.0 \times (0.71 - 0.43) = 1.40$

**Ground truth**: The correct answer is $\boxed{5}$ (the problem has a unique solution point).

*Analysis*: This example demonstrates successful diagnosis—Agent 2's critique identified a fundamental error in Agent 0's reasoning (missing constraint). Agent 0 revised the solution and fixed the error, resulting in a score increase from 0.43 to 0.71. This improvement rewards Agent 2 for providing a helpful critique that enabled better solution quality. The diagnostic reward correctly incentivizes valuable meta-evaluation.

---

### Example 2: Failed Critique (False Alarm)

**Problem (AIME 2020 Problem 8):** "A bug starts at one vertex of a cube and moves along edges. At each vertex, the bug randomly chooses one of the three adjacent edges with equal probability. What is the probability that after exactly 7 moves, the bug returns to the starting vertex?"

**Initial Solutions:** Agent 3 (Synthesizer): Uses symmetry and transition matrix. States: "Label starting vertex as distance 0. After 1 move: distance 1 (prob 1). After 2 moves: back to 0 (prob 1/3) or distance 2 (prob 2/3). Using Markov chain with states $\{0, 1, 2, 3\}$ representing distance from start, computes 7-step transition probability $P_{0,0}^7 = \frac{91}{729}$." **Answer: 91/729**

*Blind rank score:* $V_3^{\text{init}} = 0.82$ (highest, correct reasoning)

**Critique from Agent 1:** "Agent 3's answer looks clean but I think it's missing edge cases. A cube has 4 vertices at distance 3 from any starting vertex (the diagonally opposite). The transition from distance 3 back to distance 0 is impossible in one step, but Agent 3's Markov chain seems to assume all distances are reachable in one step. Did you verify the transition probabilities carefully? For instance, from distance 2 (which includes 3 vertices), what are the exact probabilities of going to each distance? I suspect there's an error in the transition matrix construction."

**Agent 3's Response:** "I appreciate the scrutiny. Let me clarify: from distance 2 (there are 3 such vertices on a cube), each vertex has 3 adjacent edges. Of these: 1 edge leads back to distance 1, and 2 edges lead to distance 3. So $P_{2\to1} = 1/3$ and $P_{2\to3} = 2/3$. From distance 3 (the single vertex diagonally opposite), all 3 edges lead to distance 2. The transition matrix is:

$$P = \begin{pmatrix} 0 & 1 & 0 & 0 \\ 1/3 & 0 & 2/3 & 0 \\ 0 & 1/3 & 0 & 2/3 \\ 0 & 0 & 1 & 0 \end{pmatrix}$$

Computing $P^7$ (which I did via diagonalization), the $(0,0)$ entry is indeed $91/729$. My answer stands."

**Final Rankings:** Agent 3's score remained high: $V_3^{\text{final}} = 0.80$ (barely changed from 0.82). Since Agent 3's score did not increase after revision, Agent 1 received zero diagnostic reward.

**Diagnostic reward for Agent 1:** $r_{\text{diag}}(1) = 5.0 \times \max(0, 0.80 - 0.82) = 0$ (no reward)

**Ground truth:** $\boxed{\dfrac{91}{729}}$ is correct.

*Analysis:* Agent 1's critique questioned valid reasoning without identifying an actual error. Agent 3 successfully defended the solution, and the score remained essentially unchanged ($0.82 \to 0.80$). Since the critiqued agent's score did not increase, Agent 1 received zero diagnostic reward, demonstrating that CoNL does not reward ineffective or misguided critiques.

---

### Example 3: Consensus Convergence

**Problem (AIME 2019 Problem 12):** "Find the number of positive integers $n$ less than 1000 for which there exists a positive real number $x$ such that $n = x\lfloor x \rfloor$."

**Initial Rankings:** Initial rankings were dispersed: $V^{\text{init}} = \{0.60, 0.55, 0.50, 0.45\}$ (Agents 0, 1, 2, 3 respectively). No clear consensus initially—all solutions seemed plausible but with different approaches and answers ranging from 496 to 512.

**Key Critique:** Agent 2 (Devil's Advocate) identified that Agent 3's solution counted certain cases twice due to an incorrect binning argument. Specifically, Agent 3 claimed that for each integer $k$, the range $k \le x < k + 1$ produces $n \in [k^2, k(k + 1))$, giving $k$ values of $n$ per bin. However, Agent 2 pointed out: "This double-counts boundaries. When $x$ is exactly an integer, $\lfloor x \rfloor = x$, so $n = x^2$, which is an integer. But your binning treats $x = k$ as belonging to bin $[k, k + 1)$ and also gives $n = k^2$ from the formula. However, the next bin $[k + 1, k + 2)$ starts at $n = k(k + 1)$, which is different from $k^2$. You need to account for overlap more carefully."

**Revised Consensus:** After revisions, the group converged on a corrected counting method:

- For $k = 1$: $n \in [1, 2)$, giving $n = 1$ (1 value)

- For $k = 2$: $n \in [4, 6)$, giving $n \in \{4, 5\}$ (2 values)

- For $k = 3$: $n \in [9, 12)$, giving $n \in \{9, 10, 11\}$ (3 values)

- ...

- For $k = 31$: $n \in [961, 992)$, giving 31 values

- For $k = 32$: $n \in [1024, \ldots)$ exceeds 1000, so partial count

Total: $1 + 2 + 3 + \cdots + 31 = \frac{31 \times 32}{2} = 496$.

**Final Rankings:** After receiving Agent 2's critique, Agent 3 revised the solution and fixed the double-counting error, arriving at the correct answer 496. Agent 3's score improved from $V_3^{\text{init}} = 0.45$ to $V_3^{\text{final}} = 0.70$. The other agents who already had correct reasoning also saw increased scores: $V^{\text{final}} = \{0.75, 0.68, 0.72, 0.70\}$, showing strong group consensus around the correct answer.

**Diagnostic reward for Agent 2**: $r_{\text{diag}}(2) = 5.0 \times (0.70 - 0.45) = 1.25$ (substantial reward for helpful critique)

**Ground truth**: $\boxed{496}$ is correct.

*Analysis*: This example illustrates consensus convergence—initially dispersed opinions coalesced around the correct answer after productive critique and revision. Agent 2 identified Agent 3's counting error, Agent 3 revised and fixed it, leading to a score increase. Agent 2 received substantial diagnostic reward for providing helpful critique that enabled solution improvement.

