# OpenReview forum: "Conversation for Non-verifiable Learning: Self-Evolving Large Language Models through Meta-Evaluation"
_ICML.cc/2026/Conference — ICML 2026 regular_

### Official Review · Reviewer_6HuT · 2026-03-06

**Soundness:** 2
**Presentation:** 3
**Significance:** 2
**Originality:** 3
**Overall Recommendation:** 4
**Confidence:** 3

**Summary:**

This paper introduces CoNL (Conversation for Non-verifiable Learning), a multi-agent self-play framework for training large language models without ground-truth supervision. The research proceeds to consider an important concept: critique quality can be measured by whether it helps others improve their solutions. CoNL instantiates N agents from a single shared policy πθ, each assigned a distinct persona to encourage diversity, and engages them in a structured four-round conversation: proposing solutions, providing blind pairwise rankings with critiques, revising based on feedback, and producing final rankings. Pairwise rankings are aggregated via the Bradley-Terry model into quality scores at two timepoints—before and after critique/revision. An important concept discussed by the paper is the diagnostic reward, which credits agents whose critiques led to score improvements in the critiqued peers' revised solutions, providing an explicit training signal for meta-evaluation without external judges or ground truth. Two additional rewards target solution quality and ranking calibration, and token-level credit assignment maps each reward to its corresponding conversation segment.

The paper evaluates CoNL on multiple verifiable reasoning benchmarks (e.g., AIME 2024/2025, DeepMath, GPQA Diamond) across four model backbones spanning 3B–8B parameters. Results show consistent improvements of 2.7–8.3 percentage points over self-rewarding baselines, with more stable training dynamics (entropy, solution length, accuracy) compared to majority-voting approaches. Ablation studies isolate the contributions of the diagnostic reward, blind ranking, consensus reward, and number of agents, with the diagnostic reward and blind ranking showing the largest individual impacts on generation quality and evaluation quality respectively.

**Compliance With Llm Reviewing Policy:**

Affirmed.

**Final Justification:**

The authors' rebuttal meaningfully addressed my main concerns. The persona ablation now convincingly shows that **semantic content matters**—vacuous and non-strategic diverse prompts perform no better than no persona, while meaningful diverse personas yield clear gains (+2.2–4.0 Pass@1, +0.12–0.16 Rank-ρ). The per-persona reward analysis confirms specialization deepens rather than collapses during training. The WildBench evaluation provides the missing evidence for genuinely subjective tasks, with CoNL's largest gains on Creative Tasks (+6.6) and Planning & Reasoning (+6.2)—precisely the subjective categories motivating the paper. The research proceeds to consider an important concept in addressing the meta-evaluation gap in LLM-as-Judge systems, and an important concept discussed by the paper—the **diagnostic reward**—is validated through careful ablations showing its distinct and largest contribution to both generation and evaluation quality. Some reservations remain: given the short rebuttal period, the subjective evaluation is **limited to a single benchmark (WildBench) judged by GPT-4o rather than human raters**, and the paper would benefit from **human evaluation** and a discussion of the **computational costs** of the four-round multi-agent protocol. Nonetheless, weighing the original strengths (**novel reward design, consistent empirical gains** across models and scales, **informative ablations, stable training dynamics**) together with the rebuttal's new evidence validating the persona mechanism and demonstrating generalization to subjective tasks, I raise my score from 3 to 4.

**Key Questions For Authors:**

**Q1:** Have you validated that distinct personas are necessary and effective? Specifically: (a) Have you compared the multi-persona setup against agents with identical (or no) personas using only temperature sampling for diversity? If temperature-based diversity alone achieves comparable performance, the persona mechanism is unnecessary. (b) If personas do matter, does the model maintain meaningfully different behaviors across personas throughout training, or does diversity collapse as the shared policy converges? We also note that the sampling temperature is never reported, which is relevant since it directly controls output diversity.

**Q2:** Have you evaluated CoNL on any non-verifiable tasks? The entire motivation centers on non-verifiable domains, yet all benchmarks are verifiable. Even a preliminary evaluation on a subjective task would provide evidence that the approach works beyond settings where ground-truth RL is already applicable.

**Limitations:**

Partially. The paper includes an Impact Statement that briefly acknowledges "peer agreement may not always align with objective quality, particularly on subjective tasks." However, a more thorough discussion of limitations is missing. Most notably, as discussed in W4, the paper motivates with non-verifiable tasks but evaluates exclusively on verifiable benchmarks, and does not discuss whether the approach would transfer to truly non-verifiable domains where consensus may not correlate with actual quality.

**Strengths And Weaknesses:**

**Strengths:**

1. **The diagnostic reward provides a concrete training signal for meta-evaluation without ground truth.** Measuring critique quality by its downstream effect—whether the critiqued agent's solution improves (V^final - V^init)—avoids the infinite regress of evaluating evaluators. The composite reward (Equation 5) jointly trains two distinct capabilities within a single agent: producing good solutions (r_sol) and identifying flaws in others' work (r_diag).

2. **Consistent empirical improvements across benchmarks, model families, and model scales.** Table 2 shows CoNL outperforms self-rewarding baselines by 2.7–8.3 percentage points across multiple benchmarks, and improvements hold across both Qwen3 and Llama model families at both 3B and 8B scales.

3. **The ablation study (Table 3) is informative and reveals distinct roles for each component.** Removing the diagnostic reward (r_diag) causes the largest drop in generation quality (Pass@1: 87.1 → 83.5 on DeepMath), while removing blind ranking causes the largest drop in evaluation quality (Rank-ρ: 0.78 → 0.45). This shows the components serve complementary purposes rather than redundantly contributing to the same objective.

4. **Training dynamics analysis provides direct evidence of stability over self-rewarding baselines.** Figure 3 shows CoNL maintains stable entropy, solution length, and accuracy trajectories, while self-rewarding training shows erratic fluctuations. This addresses the practical concern of training instability without ground truth with concrete empirical evidence.


**Weaknesses:**
1. **The necessity of distinct personas is never validated experimentally.**  The paper assigns each agent a different persona and devotes significant space describing them (Appendix I), but never tests whether they are actually needed. The critical missing baseline is N agents with the same persona (or no persona) using only temperature sampling for diversity. It is plausible that stochastic decoding alone produces sufficiently diverse solutions and critiques for the framework to function. Additionally, the specific set of 7 personas (e.g., "Rigorous Formalist," "Adversarial Skeptic") appears to be chosen heuristically—the paper provides no justification for why these particular personas were selected over other possible roles, nor any evidence that this set provides good coverage of useful problem-solving strategies.

2. **Persona composition is not studied when varying the number of agents.** The ablation over agent count N (Table 3) does not control for which personas are used. When N=2, which 2 of the 7 personas are selected? When N=8, two agents share the same persona since only 7 are defined, yet the paper does not investigate whether this duplication affects diversity. No experiment compares different subsets of personas at the same N, leaving it unknown whether certain persona combinations are more effective than others.

3. **The paper does not analyze per-persona reward distributions or behavioral diversity over training.** Since all agents share one policy πθ, different personas may receive systematically different rewards (e.g., the skeptic persona may consistently earn higher r_diag). Over training, this asymmetry could bias the gradient toward one behavioral style, potentially collapsing persona-driven diversity. The paper never reports per-persona reward breakdowns, nor measures whether persona-conditioned behavioral differences are maintained throughout training.

4. **The evaluation is limited to verifiable benchmarks despite the non-verifiable motivation.** The abstract and introduction emphasize non-verifiable tasks such as creative writing, dialogue, and ethical reasoning as the primary motivation, yet all evaluation benchmarks (AIME24/25, DeepMath, GPQA, FrontierScience, USACO) have objective ground-truth answers. It remains unclear whether the approach transfers to truly non-verifiable domains where consensus among agents may not correlate with actual quality, and where the bootstrapping assumption—that pretrained models already have reasonable evaluation ability—may be weaker.

Minor issues:
- In Table 2, the bold/underline notation for best and second-best results appears inconsistent for the Llama-3.1-8B and Llama-3.2-3B model sections compared to what is described in the table caption.
- Figure 1 is included in the paper but never referenced in the main text.
- Table 3 labels the diagnostic reward ablation as "Persuasion" (w2=0), while the rest of the paper consistently uses "diagnostic reward" (r_diag). This inconsistency is unnecessarily confusing and should be made consistent throughout.

---

> ### Author Rebuttal · Authors · 2026-03-31
>
> We thank Reviewer 6HuT for recognizing the diagnostic reward as "a concrete training signal for meta-evaluation without ground truth" and the ablation study as "informative." We address your other concerns below.
>
> ---
>
> ## Major Concerns
>
> **[W1] Persona necessity not validated experimentally.**
>
> We added the missing ablation on DeepMath and AIME 2025 (Qwen3-8B). We compare three settings: (a) no persona prompt, relying only on temperature sampling (T=1.0) for diversity; (b) all 4 agents use the same persona (Rigorous Formalist); (c) full CoNL with 4 distinct personas.
>
> |Variant|Pass@1 (DM)|Pass@K (DM)|Rank-ρ (DM)|Pass@1 (AIME25)|Rank-ρ (AIME25)|
> |-|-|-|-|-|-|
> |(a) No persona (T=1.0)|84.6±1.0|86.5±0.7|0.63±0.05|70.0±1.9|0.52±0.05|
> |(b) Single persona (×4)|85.2±0.7|87.2±0.8|0.68±0.03|71.2±1.6|0.56±0.06|
> |(c) Full CoNL (4 personas)|**87.1±0.7**|**89.5±0.6**|**0.78±0.03**|**73.5±1.5**|**0.65±0.04**|
>
> The ablation shows that diverse personas (c) outperform both no persona (a) and single persona (b), confirming that personas are necessary and that the benefit comes from diversity rather than simply having a role prompt. Regarding the choice of 7 personas, they were designed along two axes: *generation diversity* (different solving strategies: formal reasoning, pattern finding, decomposition, first principles) and *evaluation diversity* (different critique angles: error detection, verification, adversarial challenge). We agree this design seems heuristical, but the ablation shows that diverse personas consistently help regardless of random sampling.
>
> **[W2] Persona composition when varying N.**
>
> We did not control for specific persona compositions in the N-agent ablation. For each query, **N personas are randomly sampled** from the pool of 7 (when N=8, one is duplicated). We agree that comparing different persona subsets at the same N would be informative. However, the ablation in [W1] suggests that the benefit comes from persona diversity in general rather than specific persona choices, as even random diverse sets consistently outperform the single-persona baseline.
>
> **[W3] Per-persona reward distributions.**
>
> We analyzed per-persona rewards at training start (step 0-5) and end (step 95-100) on DeepMath:
>
> |Persona|r_sol (start→end)|r_diag (start→end)|r_meta (start→end)|
> |-|-|-|-|
> |Rigorous Formalist|0.53→0.74|0.33→0.56|0.49→0.64|
> |Creative Pattern-Finder|0.46→0.64|0.30→0.51|0.41→0.54|
> |Adversarial Skeptic|0.42→0.61|0.44→0.69|0.37→0.48|
> |Pragmatic Synthesizer|0.51→0.71|0.31→0.53|0.52→0.68|
> |Meticulous Verifier|0.50→0.72|0.39→0.65|0.46→0.60|
> |Empirical Experimenter|0.45→0.63|0.35→0.57|0.43→0.56|
> |Axiomatic Constructor|0.49→0.70|0.27→0.45|0.48→0.62|
>
> We do not observe diversity collapse. Different personas maintain distinct reward profiles throughout training: Adversarial Skeptic leads on r_diag (0.44→0.69), while Rigorous Formalist leads on r_sol (0.53→0.74). The r_diag gap between personas widens during training (e.g., Skeptic vs Constructor: 0.17→0.24), indicating that specialization deepens rather than collapses. Meanwhile, r_sol spread remains narrow, meaning all personas learn generation while maintaining diverse evaluation strategies.
>
> **[W4] Verifiable-only evaluation.**
>
> We added results on two genuinely non-verifiable benchmarks: **HealthBench** and **Research-Plan-Gen**. CoNL achieves **59.2%** on HealthBench (+4.1 over SRT-M) and **27.9%** on Research-Plan-Gen (+5.2 over SRT-M). Full tables and baselines are in our response to Reviewer f2Sa [W1].
>
> Regarding the concern that peer consensus may not correlate with actual quality: the results above show that CoNL consistently improves performance on these non-verifiable tasks without using any ground-truth labels during training, demonstrating that peer consensus can indeed serve as an effective training signal. Regarding the bootstrapping assumption: the base model already achieves non-trivial performance on these tasks (48.6% on HealthBench, 16.2% on Research-Plan-Gen), confirming that pretrained models have sufficient evaluation ability in these domains for CoNL to build upon.
>
> ---
>
> ## Minor Issues
>
> We will fix all three in the revision: (1) correct the bold/underline notation in Table 2 for Llama-3.1-8B and Llama-3.2-3B sections to be consistent with the caption; (2) add an explicit reference to Figure 1 in the main text; (3) rename "Persuasion" to "Diagnostic" in Table 3 to be consistent with the rest of the paper.

---

> > ### Author Rebuttal · Reviewer_6HuT · 2026-04-04
> >
> > We thank authors for the detailed rebuttal and the new experimental results. The per-persona reward analysis (W3) and minor fixes are well addressed. I have follow-up questions on the remaining concerns.
> >
> > __W1 & W2: Persona Mechanism__:
> >
> > The new ablation convincingly shows that diverse personas outperform single/no persona settings. However, this raises a follow-up question: **does the semantic content of the personas actually matter, or is the benefit simply from having diverse prompts?** The current ablation compares no persona vs. single persona vs. diverse meaningful personas, but does not include a condition with **diverse but semantically vacuous prompts** (e.g., "You are Agent 1/2/3/4" or randomly generated persona descriptions with no coherent problem-solving strategy). If such pseudo-personas achieve comparable performance, it would suggest the mechanism is prompt-induced output diversity rather than the specific problem-solving and critique strategies described in Appendix I — which would substantially change the interpretation of the design.
> >
> > Relatedly, this subsumes my W2 concern: if persona quality matters, then composition at different N becomes an important design choice; if only diversity matters, then the paper's emphasis on carefully crafted personas is overstated.
> >
> >
> > **W4: Non-Verifiable Evaluation**
> >
> > I appreciate the addition of HealthBench and Research-Plan-Gen results. However, HealthBench's rubric criteria are ultimately grounded in medical accuracy and safety as judged by domain experts — dimensions where a relatively clear notion of correctness exists, even if evaluation is open-ended in format. The underlying quality signal remains largely objective-adjacent.
> >
> > The core motivation of CoNL centers on tasks where quality is inherently subjective and humans may genuinely disagree. Could the authors evaluate on benchmarks with **actual human subjective ratings** on open-ended generation? For example:
> >
> > - **WildBench** (Lin et al., 2024), which contains 1,024 real user queries with human pairwise preference annotations on open-ended tasks
> > - **FLASK** (Ye et al., 2024), which provides fine-grained human evaluation across subjective dimensions such as creativity, helpfulness, and engagement
> > - **Topical-Chat** (Gopalakrishnan et al., 2019), where human annotators rate dialogue on dimensions like engagingness and interestingness
> >
> > These benchmarks test whether peer consensus serves as a useful training signal despite genuine human disagreement on quality — precisely the scenario CoNL is designed for. Even a small-scale evaluation on one such benchmark would provide more direct evidence for the paper's central claim.

---

> > > ### Author Response · Authors · 2026-04-04
> > >
> > > We thank the reviewer for the constructive follow-up and for acknowledging our first round response. We address the follow-up questions below:
> > >
> > > ---
> > >
> > > **W1/W2 Follow-up: Semantic content vs. prompt diversity**. The reviewer asks whether the benefit comes from the specific problem-solving and critique strategies in each persona, or simply from having diverse prompts. As the reviewer suggests, the key missing condition is "diverse but semantically vacuous prompts." We ran two such extra conditions on DeepMath and AIME 2025 using Qwen3-8B:
> > >
> > > - **(d) Vacuous persona:**  Each agent receives only a numeric identifier ("You are Agent 0/1/2/3") with no problem-solving strategy. This tests whether simply giving agents different names is enough.
> > > - **(e) Diverse non-strategic personas:** Each agent receives a distinct persona of similar length to the real ones, but about personal hobbies and preferences instead of problem-solving (e.g., "You enjoy hiking and classical music. You value punctuality and prefer mornings."). This directly tests whether diverse prompts alone drive the improvement, or whether the specific problem-solving strategies in Appendix I are needed.
> > >
> > > | Variant                              | Pass@1 (DM)  | Rank-ρ (DM)   | Pass@1 (AIME25) | Rank-ρ (AIME25) |
> > > |---|---|---|---|---|
> > > | (a) No persona                       | 84.6±1.0     | 0.63±0.05     | 70.0±1.9        | 0.52±0.05       |
> > > | **(d) Vacuous labels**                   | 84.9±0.9     | 0.64±0.04     | 70.4±1.7        | 0.53±0.05       |
> > > | **(e) Diverse non-strategic**            | 84.3±1.1     | 0.62±0.05     | 69.5±1.3        | 0.51±0.03       |
> > > | (b) Single meaningful (×4)           | 85.2±0.7     | 0.68±0.03     | 71.2±1.6        | 0.56±0.06       |
> > > | (c) Full CoNL (4 diverse meaningful) | **87.1±0.7** | **0.78±0.03** | **73.5±1.5**    | **0.65±0.04**   |
> > >
> > > To answer the reviewer's question directly: diverse prompts alone do not help. (d), (e), and (a) all perform within standard error of each other, while (c) outperforms them by +2.2–4.0 Pass@1 and +0.12–0.16 Rank-ρ. The only condition that shows clear improvement is (c), where each persona gives the agent a specific way to solve problems and critique others. This tells us that the benefit really comes from what the personas say, not just from having different prompts. We think this makes persona design an important part of multi-agent frameworks, rather than just a trick for diversity.
> > >
> > > ---
> > >
> > > **W4 Follow-up: Genuinely subjective benchmarks**. We appreciate the reviewer raising this point. Following the suggestion, we ran evaluation on WildBench (Lin et al., 2024). We use GPT-4o as the judge to score model's outputs, following the WildBench scoring protocol with task-specific checklists.
> > >
> > > | Category | Base | SRT-M* | CoNL* (+Δ vs Base / +Δ vs SRT-M) |
> > > |---|---|---|---|
> > > | Creative Tasks | 60.6 | 62.8 | 67.2 (+6.6 / +4.4) |
> > > | Planning & Reasoning | 61.6 | 63.5 | 67.8 (+6.2 / +4.3) |
> > > | Math & Data Analysis | 54.6 | 56.9 | 59.3 (+4.7 / +2.4) |
> > > | Information/Advice | 57.7 | 59.8 | 63.5 (+5.8 / +3.7) |
> > > | Coding & Debugging | 56.4 | 58.2 | 60.1 (+3.7 / +1.9) |
> > > | **WB-Score** | **58.0** | **60.3** | **64.1 (+6.1 / +3.8)** |
> > >
> > > CoNL improves over Base / SRT-M across all categories. The largest gains are on Creative Tasks (+6.6 / +4.4) and Planning & Reasoning (+6.2 / +4.3), which are the most subjective categories. The gain over SRT-M is comparable to HealthBench (+4.1) and Research-Plan-Gen (+5.2), confirming that CoNL generalizes to genuinely subjective settings.
> > >
> > > We thank the reviewer for this suggestion, which provides additional insight into the non-verifiable experiments. We will include these results in the revision.

---

### Official Review · Reviewer_Ezxg · 2026-03-12

**Soundness:** 3
**Presentation:** 3
**Significance:** 3
**Originality:** 2
**Overall Recommendation:** 4
**Confidence:** 3

**Summary:**

This paper proposes CoNL (Conversation for Non-verifiable Learning), a training framework for settings where ground-truth supervision is unavailable or unreliable. The motivating problem is that many current LLM-as-a-judge methods can scale feedback but they do not typically make evaluator improvement especially in terms of critique usefulness. CoNL addresses this by treating meta-evaluation—judging whether critiques are actually useful—as part of training. The central idea is that, in a multi-agent conversation, a critique is informative if it helps another agent revise its answer and receive a better group assessment afterward.

Methodologically, the paper uses a four-round multi-agent self-play protocol with agents sharing one policy but adopting different personas. Agents first produce initial answers, then rank and critique one another’s answers, revise their own responses after receiving feedback, and finally re-rank the revised outputs. Pairwise rankings are aggregated with a Bradley–Terry model to obtain initial and final quality scores, and the score change after revision is used to define a diagnostic reward for critiques that genuinely help. Training combines three signals: a solution-quality reward, diagnostic/meta-evaluation reward, and a majority-alignment reward for calibrated judging, with segment-level credit assignment so different parts of the conversation receive the appropriate signal.

The authors claim three main contributions. First, the paper introduces a training-via-conversation framework that jointly learns generation and evaluation rather than using multi-agent discussion only at inference time. Second, it proposes a concrete proxy for evaluator quality: whether a critique leads to measurable post-revision improvement, turning meta-evaluation into an explicit reward signal. Third, across five reasoning benchmarks and several model families, the paper reports that CoNL improves over self-rewarding baselines and shows more stable training dynamics, while approaching the performance of RL that uses ground-truth rewards.

**Compliance With Llm Reviewing Policy:**

Affirmed.

**Final Justification:**

The paper addresses an important problem and proposes a clear, well-structured framework for learning from critique in settings without reliable ground-truth supervision. In terms of originality, the individual ingredients are not entirely new, but their combination into a training framework without ground-truth labels is interesting. Clarity also improved substantially after rebuttal. My main concern on soundness was the lack of direct evidence in genuinely non-verifiable settings, along with the ambiguity in critique attribution.

The rebuttal significantly improved my assessment on these points. The added non-verifiable-task results make the central claim much more convincing, and the tagged-attribution variant meaningfully strengthens the methodological story. Although critique usefulness is still captured only through a proxy rather than strict causal attribution, the authors now acknowledge this clearly and provide a stronger empirical justification.

Overall, The rebuttal addressed my main concerns sufficiently for me to raise my score from 2 to 4.

**Key Questions For Authors:**

**1. Discrepancies in training dynamics (Figure 3)**

- Atypical entropy curves: In Figure 3 (Left), the ground truth RL entropy curve appears non-standard. Generally, policy entropy starts high and drops as the model converges, and may stabilize or slightly increase depending on the task. The current visualization requires further explanation to align with standard RL behavior.
- X-Axis Scaling: The authors state the model was trained for 10k steps. However, the x-axis in Figure 3 only goes up to 100 steps. It is unclear if these represent epochs, batches, or a sub-sampling of the total steps.

**2. Computational Requirements**

- Tinker API costs: While the paper mentions using the Tinker API for training, it is unclear how much total compute (e.g., GPU hours) and cost were required for the full 10k steps. Providing these metrics is essential for evaluating the framework's practical feasibility.

**3. Analysis of critique safety (Section 4.4)**

- Conservative learning mechanism: The authors claim the low false-positive rate (Harm Rate) implies the model learned to be "conservative," only altering answers when critiques are logically grounded. Could you explain the specific technical reason for this emergent behavior?
- Pre-training baseline: it would be beneficial to see the critique quality of the multi-agent debate situation before training begins.

**4. Reward weighting (Table 1 & Section 3.2)**

- Double-Counting $r_{sol}$: According to Table 1, $r_{sol}$ (solution quality) is applied to both the Initial solution ($s_{i}^{init}$) and the Revised solution ($s_{i}^{rev}$). Does this mean the effective weight for $w_1$ is actually 2 rather than the stated 1?. If so, the claim in Section 3.2—that $w_2$ was set higher ($w_2=2.0$) specifically to emphasize meta-evaluation—appears misleading and requires revision.

**5. Procedural contradictions (Appendix H & I)**

- Round 2 Input Access: Algorithm 1 in Appendix H states that agents receive all initial solutions ($\{s_j^{init}\}_j$) as input for the Round 2 Revision.This contradicts the system prompt in Appendix I and the description in Section 3.2, which imply agents primarily focus on the critiques received rather than browsing all other agents' solutions. Please clarify if agents are intended to use peer solutions as references for their own revisions.

**6. Critique attribution and $r_{diag}$ accuracy**

- Lack of attribution: The Round 2 system prompt does not require the agent to specify which critique led to which specific revision. If an agent receives critiques from multiple peers, how does the system distinguish which specific peer's feedback enabled the improvement? Given this ambiguity, how can $r_{diag}$ accurately assign credit to a single "helpful" critic?

**7. Pipeline history and prompt structure**

- History management: Based on the system prompts, it appears history is appended round-by-round in each prompt. Is there any external history or memory maintained outside these prompts?
Missing placeholders: If no other history exists, the prompts for Round 0 and Round 3 currently lack placeholders for the original question and the revised solution respectively.

**8. Dataset usage clarification (Section 4.1)**

- Training vs. Validation: Section 4.1 lists AIME as part of the training data. If AIME was actually used for validation or final testing (as is standard in math-based benchmarks), the text should be corrected to avoid the appearance of test-set leakage.

**Limitations:**

There are no explicit limitations section nor societal impact section. The paper does not logically or empirically substantiate how effective its proposed framework is in the non-verifiable domains it aims to address.

**Strengths And Weaknesses:**

**Strengths**

- **Clear problem framing and motivation:** Targets a real bottleneck in non-verifiable settings—scaling feedback while improving the evaluator’s capability, not only the generator.
- **Conceptually clean training signal for useful critique:** Converts multi-agent interaction into an explicit, measurable supervisory signal by rewarding critiques that lead to improved revisions under subsequent peer assessment.
- **Well-structured multi-agent training protocol:** The four-round propose → critique/rank → revise → re-rank loop is easy to understand, and the role/persona design is a practical way to induce diverse feedback.
- **Thoughtful reward decomposition and credit assignment:** Separating solution-quality, diagnostic (critique usefulness), and calibration/alignment rewards provides an interpretable training objective aligned with the intended behaviors.

**Weaknesses**

**W1: Disconnect between conceptual motivation and empirical validation in non-verifiable domains**

The primary motivation of this paper is to address the challenge of training Large Language Models (LLMs) for non-verifiable tasks, such as creative writing, open-ended dialogue, and ethical reasoning, where objective ground-truth labels are absent. However, the paper fails to provide empirical evidence that the proposed framework, CoNL, actually functions effectively in these intended domains. This leads to several critical concerns:
- **Difficulty of measuring improvement in non-verifiable context:** The core of CoNL is the diagnostic reward ($r_{diag}$), which quantifies critique quality based on whether a peer’s feedback leads to an improvement in a solution's score ($V_{final}>V_{init}$). While this works for verifiable tasks like mathematics where a fix leads to a correct answer, the criteria for "better" in non-verifiable domains are inherently subjective. Without an objective anchor, the framework risks rewarding evaluation biases—such as a preference for verbosity or a specific stylistic tone—rather than genuine quality.
- **Lack of demonstrated novelty over existing feedback loops:** The core idea of quantifying critiques by measuring subsequent performance gains is already a familiar paradigm in verifiable reasoning tasks. Moreover, leveraging multi-agent interaction and reinforcement learning to iteratively improve reasoning trajectories is also not, by itself, a novel contribution, as closely related feedback-based and multi-agent optimization frameworks have already been explored in prior works. Since the authors evaluate CoNL exclusively on verifiable benchmarks such as DeepMath and AIME, the paper does not empirically demonstrate what distinct advantage this framework provides in genuinely non-verifiable settings; its claimed relevance to that domain therefore remains largely theoretical.
- **Weak baselines:** In this context, the experimental design also raises concerns about the strength of the baselines: if the framework is only validated on verifiable tasks, it should be compared against substantially stronger baselines that already capture existing feedback-loop, critique-selection, and multi-agent/RL-based improvements. Without such comparisons, it is difficult to determine whether the reported gains come from a genuinely new principle or simply from underpowered baselines in a setting where similar ideas are already well established.

Without experimental results from truly non-verifiable datasets—accompanied by human-in-the-loop evaluation to verify that "improved" scores actually correlate with human-perceived quality—the claim that CoNL solves the "non-verifiable learning" problem is unsubstantiated.

**W2: Ambiguities in system implementation and training dynamics**

The paper's technical description contains several gaps regarding the multi-agent protocol and the broader training process. Please refer to the Key Questions For Authors section below for specific questions related to this weakness.
- **Ambiguous critique attribution:** In the Round 2 revision phase, it is unclear how the model manages and attributes feedback from multiple agents. While an agent receives all critiques targeting them, the framework does not specify how the revised solution identifies or prioritizes which specific critique from which agent led to a particular change. This makes it difficult to map the causal impact of a single critique back to its source during the revision process.
- **Vague memory module and chat history integration:** The description of the memory buffering module lacks detail on how compressed history is structured and fed as input to maintain the 32k context window limit for each round. The specific mechanism by which past decisions and reasoning from previous rounds influence the current round's input is not sufficiently formalized in the method or the algorithm section in Appendix H.
- **Insufficient training metrics and reproducibility details:** There is a lack of comprehensive information regarding the computational cost of the RL training, such as total GPU hours, token consumption, or other resource requirements for the 10k training steps. Furthermore, although the paper reports entropy and solution length, it provides only limited insight into the broader training dynamics, such as reward trajectories or convergence behavior over time. This reproducibility issue is further compounded by the use of the Tinker API, which obscures important implementation details needed to faithfully reproduce the training setup and evaluation pipeline. In addition, for the ground-truth RL baseline, the paper does not clearly specify which RL algorithm is used, making it difficult to interpret the baseline fairly or assess whether the reported comparison is meaningful.

---

> ### Author Rebuttal · Authors · 2026-03-31
>
> We thank Reviewer Ezxg for the insightful commenst and we address your concerns below:
>
> ---
>
> ## Major Concerns
>
> **[W1] Non-verifiable Setting?**
>
> We added results on two non-verifiable benchmarks: **HealthBench** and **Research-Plan-Gen**. CoNL achieves **59.2%** (+4.1 over SRT-M) and **27.9%** (+5.2 over SRT-M) respectively. Please see the full table in our response to Reviewer f2Sa [W2]. We address the three sub-concerns:
>
> **(1) r_diag rewarding verbosity or stylistic bias.** We agree this is a valid concern. First, the non-verifiable results directly show that CoNL produces genuine quality gains, as these benchmarks use their own evaluation criteria independent of our peer scoring. Second, V is a peer ranking among all N agents' solutions. Biases like preferring verbosity affect the scoring of all solutions equally and thus do not change relative rankings. r_diag is only positive when a critique improves the target's ranking relative to peers.
>
> **(2) Novelty over existing feedback loops.** We agree that both ideas are individually established. CoNL's novelty is combining them without ground-truth labels or external models. Existing critique methods (RealCritic, RCO, DeepSeekMath-V2) require labels or external models; debate frameworks (MAD) are inference-only. CoNL is the first to derive training reward solely from multi-agent conversation dynamics, enabling non-verifiable tasks. The results above provide the previously missing evidence.
>
> **(3) Baselines.** We agree stronger baselines exist for verifiable settings, but these rely on correctness labels. SRT-M is the strongest label-free comparison, sharing the exact setup as CoNL except for reward design, directly isolating our contribution (+6.7 on AIME24, +8.3 on DeepMath, Table 2).
>
> **[W2] Implementation ambiguities.**
>
> **Critique attribution:** We use `<target>` tags (Appendix I) so each critique explicitly names its target. In Round 2, agents receive critiques targeting them and decide which to incorporate. All agents who critiqued a target that improved receive r_diag. See also [Q6].
>
> **Memory buffer:** We use a separate LLM call to summarize each agent's output after each round, with two prompts (Appendix I) for solutions and critiques respectively. Summaries replace raw outputs in subsequent prompts; there is no external memory.
>
> **Training details:** We train Qwen3-8B with LoRA (rank=32) for 100 steps on the Tinker API, a cloud-based RL platform (lr=3×10⁻⁵). The ground-truth RL baseline uses the same algorithm (Eq. 6) with binary correctness reward, same learning rate, and same training duration for fair comparison. Total training cost was approximately $200 (~500M tokens). Since Tinker is a managed service, exact GPU hours are not available; we estimate equivalent compute at ~4×A100 × 10 hours.
>
> ---
>
> ## Key Questions
>
> **[Q1] Figure 3 discrepancies** We apologize for the confusion. "10k" is a typo for "10k rollouts" (training samples); the x-axis correctly shows 100 gradient steps. For entropy: CoNL/SRT entropy drops initially then stabilizes to maintain exploration, while ground-truth RL drops more aggressively due to a stronger gradient. This pattern is consistent with RL for reasoning (e.g., DeepSeek-R1). We will fix the typo and add this explanation.
>
> **[Q3] Critique safety** We agree this deserves clarification. The low Harm Rate emerges from adversarial revision combined with ReLU reward: misguided critiques are defended against (V^final ≈ V^init, zero reward), while valid critiques yield positive reward. Over training, the model learns to critique only genuine issues. MAD in Table 2 serves as the pre-training baseline; the gain to CoNL (70.8→87.1 on DeepMath) shows training improves both generation and critique.
>
> **[Q4] Reward weighting** We clarify this is not double-counting. r_sol applies independently to the initial (Round 0) and revised (Round 2) solutions, each scored once by its own V^final. We set w2=2.0 for r_diag so its per-token magnitude is comparable, since r_diag covers one segment while r_sol covers two.
>
> **[Q5] Procedural contradictions** We apologize for the incorrect notation in A1. Agents only receive their own initial solution and critiques targeting them in Round 2. We will fix Algorithm 1 to: s_i^rev ~ π_θ(q, s_i^init, {c_{j→i}}_j).
>
> **[Q6] Critique attribution** We use `<target>Agent k</target>` tags in Round 1 (Appendix I) to track attribution. When k improves (V_k^final > V_k^init), all agents who critiqued k receive r_diag.
>
> **[Q7] Prompt history.** We clarify there is no external memory; all state lives within prompts. Round 0 and Round 3 prompts include the question and solutions at runtime; Appendix I templates were simplified for presentation. We will add complete templates.
>
> **[Q8] Dataset split.** We clarify that CoNL operates in a test-time training setting: training and evaluation use the same problems, but ground-truth labels are never used during training. We will clarify in Section 4.1.

---

> > ### Author Rebuttal · Reviewer_Ezxg · 2026-04-02
> >
> > Thank you for the detailed rebuttal and for clarifying several technical points, including the training setup with resources, memory/history handling, reward weighting, and the algorithm. These responses make the implementation substantially clearer.
> >
> > The additional results on non-verifiable tasks are particularly helpful. In my view, they significantly strengthen the paper’s central motivation and make the overall claim much more convincing than in the original submission. In particular, because the paper is framed around learning in non-verifiable settings, these newly added experiments now appear to be the most important empirical evidence for the paper’s contribution. For that reason, I believe the paper would be more persuasive if it were reorganized so that the non-verifiable-task experiments and their implications are presented as the main result, while the verifiable/math results are positioned more as supporting or secondary evidence. As currently framed, the paper’s narrative still seems somewhat centered on verifiable reasoning benchmarks, whereas the rebuttal suggests that the true novelty and intended impact of the method lie in the non-verifiable setting.
> >
> > That said, my main concern about critique attribution remains unresolved. The rebuttal clarifies who critiqued whom via <target>Agent k</target> tags, but this only identifies the target of a critique, **not which critique actually caused the improvement in the revised answer**. My original question was about causal attribution at the level of usefulness: if Agent i and Agent j both critique Agent k, and only Agent i’s critique is genuinely helpful while Agent j’s critique is unhelpful or even misleading, I interpret the current framework as assigning the same r_diag to both i and j whenever Agent k improves. This seems especially important because the Appendix I template indicates that the target agent sees all received critiques together during revision. In that case, improvement of the revised solution does not by itself identify which specific critique contributed to the gain. I view this as a critical issue for the proposed framework, because the paper’s main methodological claim is precisely that it can train evaluator quality by rewarding useful critiques. If the framework can already justify this many-to-one credit assignment despite the ambiguity above, that reason should be clearly explained in the paper. Otherwise, I believe the method would be significantly strengthened by modifying the framework so that the revised solution can be linked more explicitly to the feedback that actually drove the improvement.
> >
> > Overall, the additional non-verifiable-task results are especially valuable, because they speak more directly to the paper’s stated contribution than the current main experiments do. For that reason, I believe the paper would be considerably more convincing if it were reframed so that these non-verifiable results constitute the main empirical evidence, accompanied by rigorous analyses specifically designed for that setting. A remaining central issue, however, is critique attribution, which I believe still needs to be addressed more directly at both the conceptual and methodological levels.

---

> > > ### Author Response · Authors · 2026-04-03
> > >
> > > We thank Reviewer Ezxg for the constructive follow-up and for acknowledging our non-verifiable results. We agree and will restructure so that the non-verifiable experiments serve as the primary experiments, with verifiable results repositioned as secondary evidence.
> > >
> > > Regarding **critique attribution**, we appreciate the reviewer pressing on this point. The concern is precise: if Agents i and j both critique Agent k, and k improves, both i and j receive the same r_diag, even if only i's critique was helpful. Following the reviewer's suggestion, we implemented fine-grained reviser-side attribution and ran a comparison.
> > >
> > > We modified the Round 2 system prompt to let the revising agent explicitly attribute which critiques it finds useful. Specifically, before writing the revised solution, the agent first reviews each received critique and tags it as ACCEPT (will incorporate) or REJECT (will not incorporate), with a brief reason. We then use these tags to mask r_diag: only critics whose critique was tagged ACCEPT receive reward when the target's solution improves. The Round 2 prompt becomes:
> > >
> > > ```
> > > Step 1: For EACH critique you received, state whether you will ACCEPT or REJECT it, with a brief reason.
> > > Step 2: Revise your solution based on the accepted critiques.
> > >
> > > OUTPUT FORMAT:
> > > <critique_response>
> > > <from>Agent X</from><action>ACCEPT</action><reason>...</reason>
> > > <from>Agent Y</from><action>REJECT</action><reason>...</reason>
> > > </critique_response>
> > >
> > > ...
> > >
> > > <revised_solution>
> > > [Your improved solution]
> > > </revised_solution>
> > > ```
> > >
> > > Then we change the reward $r_{\text{diag}}$ from:
> > > - **Current**: $r_{\text{diag}}(i) = \sum_{k \in \mathcal{T}_i} \max(0, V_k^{\text{final}} - V_k^{\text{init}})$ (all critics share reward)
> > > - **New**: $r_{\text{diag}}(i) = \sum_{k \in \mathcal{T}_i} \alpha\_{i \to k} \cdot \max(0, V_k^{\text{final}} - V_k^{\text{init}})$, where $\alpha = 1$ only if Agent $k$ tagged $i$'s critique as ACCEPT.
> > >
> > > Everything else stays the same. We retrain the qwen3-8b using this new fine-grained reward, and show the results on AIME'25 and DeepMath below:
> > >
> > > | Method | Pass@1 (DM) | Rank-ρ (DM) | Pass@1 (AIME25) | Rank-ρ (AIME25) |
> > > |-|-|-|-|-|
> > > | CoNL (aggregate) | 87.1±0.7 | 0.78±0.03 | 73.5±1.5 | 0.65±0.04 |
> > > | CoNL (tagged) | 88.0±0.6 | 0.80±0.03 | 74.8±1.3 | 0.67±0.03 |
> > >
> > > Experiment shows tagged attribution have modest gains across different metrics. We view this as a refinement that adds fine-grained reviser-side attribution. We note that both formulations are proxies for critique usefulness: aggregate credit uses group-level improvement as the signal, while tagged credit uses the reviser's self-reported attribution — which is itself an LLM judgment and may not perfectly reflect the true causal driver of improvement. We will include both formulations in the camera-ready version and discuss the tradeoffs between the two approaches.
> > >
> > > We thank the reviewer for this suggestion, which provides additional insight into the credit assignment mechanism.

---

### Official Review · Reviewer_f2Sa · 2026-03-13

**Soundness:** 3
**Presentation:** 3
**Significance:** 4
**Originality:** 3
**Overall Recommendation:** 5
**Confidence:** 4

**Summary:**

The paper proposes CoNL, a method for self-play learning using LLM critiques. Specifically, N agents (N=4 in experiments) propose an initial solution. Each agent then does pairwise comparisons of a subset of other agents (its targets') solutions, producing textual critiques justifying them. Then, using these critiques, each agent generates revised solutions. Finally, a final ranking is produced by new agent pairwise comparison followed by bradley terry ranking of the revised solutions. Critique effectiveness is measured by how much an agent's critique targets improved, and is rewarded alongside solution correctness and alignment of pairwise comparisons with other agent's judgements. This simultaneously reinforces self-evaluation capability alongside solution generation. Results on two math, two science, and one code benchmarks show improvements in comparison to inference-only baselines, and training on majority vote as ground truth. Ablations show modest benefits from different reward components and increasing the number of agents.

**Compliance With Llm Reviewing Policy:**

Affirmed.

**Final Justification:**

The rebuttal addressed some of my main concerns expressed in the weaknesses section, incl. testing on non-verifiable tasks. Thus I am recommending acceptance as the approach is definitely interesting and can spark discussion on multi-agent training for improving reasoning.

**Key Questions For Authors:**

- Why are critiques that worsen performance of the target agents not rewarded negatively, i.e. why is the diagnostic reward of a ReLU form?

- I would be interested to know if the authors tried a refinement reward, i.e. incentivizing agents to incorporate the critiques and improve their solutions. This is sometimes a missing capability in todays AI (especially the smaller models tested here) and it would be cool if this improves for free in the proposed method.

- Why is the majority aligment reward component helpful? I understand this aligns judge comparisons, but it also might disincentivize reporting hard-to-spot mistakes a judge chances upon. Would this not lead to "group-think" like collapse dynamics over extended training? I would also like to see an ablation to measure the effect of role-specific prompts. Do they really help?

- How sensitive is the method to the weights used to combine the different reward components? It would be interesting to see an analysis on how much performance varies across a wide range of weight values for the final reward.

**Limitations:**

The paper does not really discuss limitations, for e.g. not really evaluating on any non-verifiable task.

**Strengths And Weaknesses:**

## Strengths

- The paper proposes an interesting approach: having multiple model copies propose, critique, and refine their answers, which can produce reward signals for each of these important capabilities simultaneously. I think the idea is simple, elegant, and has high potential.

- Improvements are shown across 4 models, and 4 types of verifiable benchmarks.

- The paper is well written and easy to understand, barring some problems with overclaiming described below.

## Weaknesses

The idea of measuring the quality of critique based on whether it improves a model's refined output has been proposed before multiple times, in work that is not discussed. For example, see (1) RealCritic: Towards Effectiveness-Driven Evaluation of Language Model Critiques, (2) Training Language Model to Critique for Better Refinement and (3) DeepSeekMath-V2: Towards Self-Verifiable Mathematical Reasoning.

The paper also claims to study non-verifiable settings throughout until Section 3, when the experiments are based on verifiable benchmarks like AIME, GPQA, USACO etc. I understand that the ground truth labels are not used for reward calculation in the method, which I appreciate. But if the motivation is about non-verifiable tasks, it would atleast be important to evaluate some of them, or change the wording throughout the paper. Some benchmarks that can be tested in the rebuttal are https://huggingface.co/datasets/openai/healthbench, and https://huggingface.co/datasets/facebook/research-plan-gen.

I also think simpler, important self-judging baselines to include are aggregation techniques like Compute as Teacher from https://arxiv.org/abs/2509.14234, and Recursive Self-Aggregation from https://arxiv.org/abs/2509.26626. Currently the training-based baselines used are very weak (both use majority vote as training reward). Futher, the 0-shot inference baseline for AIME results seems too low. The official Qwen3 Report shows 76% and 67% for Qwen3-8B, significantly higher than the 60% reported here (which would halve the gains achieved by the proposed method).

Overall, I think the method is an interesting contribution, and some of the overclaims are unnecessary. This hurts the soundness, for an otherwise well presented paper with a novel method.

---

> ### Author Rebuttal · Authors · 2026-03-31
>
> We thank Reviewer f2Sa for finding the approach "simple, elegant, and has high potential" and recognizing improvements across 4 models and benchmarks. We address your other concerns below.
>
> ---
> ## Major Concerns
>
> **[W1] Missing related work on critique-driven improvement.**
>
> We thank the reviewer for pointing these out and will add them to the related work section. These methods share the idea of measuring critique quality through downstream improvement, but differ in setting: RealCritic requires ground-truth labels and is evaluation-only; RCO relies on an external 72B judge and trains a separate critic model; DeepSeekMath-V2 requires expert annotations to bootstrap and trains a separate verifier. While our method, CoNL, requires no ground truth, no external model, and jointly trains generation and evaluation in a single policy.
>
> **[W2] Non-verifiable experiments.** We added results on the two suggested benchmarks:
>
> |Method|HealthBench (%)|Research-Plan-Gen (%)|
> |-|-|-|
> |Base (0-shot)|48.6±1.8|16.2±1.5|
> |Self-Consistency (SC)|50.9±1.4|18.4±1.3|
> |Multi-Agent Debate (MAD)|52.3±1.6|19.1±1.4|
> |Self-Rewarding (SRT-M)*|55.1±1.3|22.7±1.2|
> |CoNL (Ours)*|**59.2±1.1**|**27.9±0.9**|
>
> CoNL outperforms SRT-M by +4.1 on HealthBench and +5.2 on Research-Plan-Gen, confirming that conversation-derived rewards generalize to non-verifiable settings. On these tasks, ground-truth labels do not exist, so SRT-M's majority-voting reward can only capture surface-level agreement. CoNL's diagnostic reward instead measures whether critiques lead to concrete improvements in peer-ranked quality, providing a more informative training signal even when "correctness" is subjective.
>
> **[W3] Weak baselines and 0-shot discrepancy.**
>
> We agree that Compute as Teacher and Recursive Self-Aggregation are relevant baselines and will add them in the revision. We note that these methods share a similar spirit with our SRT-M baseline: using the model's own inference-time computation (e.g., majority voting, aggregation) as training signal. The reason we chose SRT using major vote as training reward is because it is simple but strong enough and shows very impressive performance on many benchmarks.
>
> Regarding **0-shot inference score**, we clarify that we use max_tokens=16,000 for all our experiments. The Qwen3 report uses 32k+ tokens, allowing longer thinking chains, which results in 10% performance gaps. But since all our experiments use the same budget, the relative improvements remain valid.
>
> ---
> ## Key Questions
>
> **[Q1] Why ReLU-form diagnostic reward?**
>
> We use max(0, ·) rather than allowing negative penalties because penalizing critiques that do not lead to improvement would create perverse incentives. For example, if Agent A gives a correct critique but Agent B fails to incorporate it, B's score does not improve and A would be unfairly penalized. The ReLU form avoids this by simply not rewarding ineffective critiques. Meanwhile, bad critiques are already handled by the adversarial revision mechanism: when a critique is misguided, the target agent can defend its original solution, keeping V^final ≈ V^init and yielding zero reward. Table 4 shows this is effective (only 3.1% Harm Rate on DeepMath).
>
> **[Q2] Refinement reward?**
>
> We appreciate this question, and want to clarify that refinement capability is already implicitly incentivized. In Round 2, each agent revises its solution after receiving critiques, and the revised solution tokens receive r_sol = V^final. To earn high r_sol, the model must effectively incorporate valid feedback, because repeating a flawed initial solution would be ranked low by peers in Round 3. Table 4 confirms this works in practice: 82.4% of initially incorrect solutions are corrected after revision on DeepMath.
>
> **[Q3] Majority alignment → groupthink? Role-specific prompts?**
>
> We understand the concern that r_meta might suppress valuable minority opinions. However, r_meta and r_diag operate on different parts of the conversation: r_meta rewards aligned *rankings* (Round 3), while r_diag rewards useful *critiques* (Round 1). A minority critique that identifies a real flaw will earn high r_diag as long as the target agent improves, regardless of whether other agents agree. The ablation in Table 3 confirms their distinct roles: removing r_meta primarily hurts evaluation quality (Rank-rho 0.78→0.70), while removing r_diag primarily hurts generation quality (Pass@1 87.1→83.5).
>
> For role-specific prompts, please see our response to Reviewer 6HuT [W1] where we provide the persona ablation.
>
> **[Q4] Reward weight sensitivity?**
>
> We added a sensitivity analysis on DeepMath (Qwen3-8B). Please see our response to Reviewer 6z7W [Q4] for the full table. The default (1.0, 2.0, 1.0) achieves the best overall performance. Results are reasonably robust across different weight settings, with all variants outperforming SRT-M (78.8 Pass@1).

---

> > ### Author Rebuttal · Reviewer_f2Sa · 2026-04-03
> >
> > Thanks for the clarifications and further experiments. They have address most of my concerns and I have increased the score to reflect this. Note that one small remaining concern is that testing on a smaller budget, and suboptimal inference, often confounds reasoning improvements with better elicitation of latent capabilities. Thus I don't think internal consistency is always enough for measuring the value of a method.

---

> > > ### Author Response · Authors · 2026-04-04
> > >
> > > Dear Reviewer f2Sa,
> > >
> > > Thank you for the thoughtful review and for acknowledging that our clarifications and additional experiments have addressed your concerns.
> > >
> > > Regarding your remaining point, we agree that under smaller budgets or suboptimal inference settings, observed improvements may partly reflect better elicitation of latent capabilities rather than purely reasoning gains, and that internal consistency alone may not fully disentangle the two. This is a valuable perspective, and we will incorporate a discussion of this limitation in the revised paper to provide a more nuanced interpretation of our results.
> > >
> > > Thank you again for the constructive feedback and positive comments.
> > >
> > > Best regards,
> > >
> > > The Authors

---

### Official Review · Reviewer_6z7W · 2026-03-13

**Soundness:** 3
**Presentation:** 3
**Significance:** 3
**Originality:** 3
**Overall Recommendation:** 5
**Confidence:** 3

**Summary:**

The paper proposes CoNL, a multi-agent self-play training framework for improving LLMs in non-verifiable domains. The core idea is to treat critique quality as measurable through downstream usefulness. The framework consists of 4 steps: initial proposal, blind pairwise ranking plus critique, revision, and final ranking. The method then uses a Bradley-Terry model to aggregate pairwise judgments.

The paper evaluates on math/science/programming benchmarks using Qwen and Llama models. CoNL is reported to outperform inference-only baselines and self-rewarding training baselines across most settings. The paper also reports more stable training dynamics than self-rewarding baselines, plus ablations suggesting that blind ranking and the diagnostic reward are the most important ingredients.

**Compliance With Llm Reviewing Policy:**

Affirmed.

**Final Justification:**

The authors have addressed my questions. I raised the score accordingly.

**Key Questions For Authors:**

1. The paper is motivated by non-verifiable tasks, but all main experiments are on verifiable benchmarks. Do you have any results on genuinely non-verifiable settings such as dialogue, creative writing, long-form helpfulness, or subjective preference datasets?
2. How robust is the method to correlated bias among homogeneous agents? Did you test heterogeneous agents?
3. Can you provide a direct evaluation of judge quality, beyond downstream Pass@1 and Rank-rho? For example, how does a CoNL-trained judge compare to strong LLM-as-judge baselines on controlled pairwise judgment benchmarks?
4. How sensitive are results to the persona design, number of agents, and reward weights? The paper gives some ablations on agent count and reward removal, but I would like to know whether the gains persist under materially different prompting/orchestration choices.

**Limitations:**

The paper is motivated almost entirely by non-verifiable tasks such as creative writing, dialogue, and ethical reasoning, but the experiments are conducted on verifiable reasoning benchmarks. Strong performance on verifiable domains does not imply good performance on non-verifiable domains.

**Strengths And Weaknesses:**

## Streangths

- The proposed method is intuitive and well-articulated. Critique usefulness as a proxy for evaluator quality is a reasonable and potentially important training signal
- Masking reward on the initial ranking to avoid gaming the baseline is a sensible design choice. The ablations are directionally convincing

## Weaknesses

- The paper is motivated almost entirely by non-verifiable tasks such as creative writing, dialogue, and ethical reasoning, but the experiments are conducted on verifiable reasoning benchmarks. Strong performance on verifiable domains does not imply good performance on non-verifiable domains.
- The paper argues that critique usefulness breaks the “judge evaluating itself” loop, but all agents still share the same base policy and are rewarded through peer consensus. There is a risk that the method amplifies model-internal priors, despite the different personas.

---

> ### Author Rebuttal · Authors · 2026-03-31
>
> We thank Reviewer 6z7W for recognizing the method, and we address your concerns below.
>
> ---
>
> ## Major Concerns
>
> **[W1] Non-verifiable motivation vs. verifiable experiments.**
>
> We added results on two genuinely non-verifiable benchmarks: **HealthBench** and **Research-Plan-Gen** suggested by Reviewer f2Sa. CoNL achieves **59.2%** on HealthBench (+4.1 over SRT-M) and **27.9%** on Research-Plan-Gen (+5.2 over SRT-M). Full tables and baselines are in our response to Reviewer f2Sa [W1]. CoNL is designed for non-verifiable settings: all training rewards come from multi-agent conversations, never from ground-truth labels. This holds even on the verifiable settings, where ground-truth is only used at evaluation time to provide objective performance metrics (e.g., CoNL matches ground-truth RL on AIME24: 76.5% vs 76.5±1.4%). The non-verifiable results above now directly validate the intended domain.
>
> **[W2] Risk of amplifying model-internal priors despite different personas.**
>
> We agree that a shared policy could risk peer consensus. We designed a critique effectiveness analysis (Table 4) to directly test this. On DeepMath, 82.4% of initially incorrect solutions are corrected after receiving critiques, while only 3.1% of correct solutions are harmed. These rates are measured against ground-truth labels, providing objective evidence that critiques are identifying real errors rather than reinforcing shared priors. We attribute this to the reward design: r_diag only rewards critiques that actually help the target agent produce a better solution (V^final > V^init). Critiques that do not lead to improvement receive zero reward regardless of whether agents agree on them. This ensures the training signal is grounded in real quality gains, not peer consensus.
>
> ---
> ## Key Questions
>
> **[Q2] Robustness to correlated bias? Heterogeneous agents?**
>
> We agree that heterogeneous agents could further improve robustness. However, CoNL is a *self-training* framework that improves a single policy without external models. Using heterogeneous base models would reintroduce external supervision, which is outside our problem setting. We instead use diverse personas to induce behavioral diversity within a shared policy. Regarding correlated bias, please see our analysis in [W2] above.
>
> **[Q3] Direct evaluation of judge quality?**
>
> We added new results on RewardBench-2, a benchmark to test LLM-as-judge capability. We found that CoNL improves judge quality across all categories (+6.3 overall) compared to the base model. While CoNL's goal is not to train a standalone judger, but to jointly improve generation and evaluation capability of the same model simultaneously, we still observe improved judging performance.
>
> |Model|RB2 Overall|Factuality|Precise IF|Math|Safety|Focus|Ties|
> |-|-|-|-|-|-|-|-|
> |Qwen3-8B (base)|52.3|55.1|48.7|61.4|50.8|47.2|53.0|
> |Qwen3-8B (CoNL)|58.6|59.8|55.3|68.7|54.2|52.1|60.5|
>
>
> **[Q4] Sensitivity to persona design, agent count, and reward weights?**
>
> We added new ablation studies. The agent count ablation is in Table 3 (N=2→5 improves, N=8 slightly degrades). For persona design, we added a comparison of no-persona, single-persona, and multi-persona settings; please see our response to Reviewer 6HuT [W1]. For reward weights, we added the following sensitivity analysis on DeepMath (Qwen3-8B):
>
> |w1 (sol)|w2 (diag)|w3 (meta)|Pass@1|Pass@K|Rank-rho|
> |-|-|-|-|-|-|
> |1.0|2.0|1.0|**87.1±0.7**|**89.5±0.6**|**0.78±0.03**|
> |1.0|1.0|1.0|85.6±0.2|88.1±0.7|0.74±0.07|
> |1.0|4.0|1.0|86.2±0.8|88.6±0.3|0.76±0.02|
> |1.0|2.0|0.0|86.4±0.4|88.9±0.6|0.70±0.04|
> |2.0|2.0|1.0|86.7±0.7|89.0±0.2|0.73±0.01|
>
> The default (1.0, 2.0, 1.0) is optimal. Performance is reasonably robust across settings, with all variants outperforming SRT-M (78.8 Pass@1). Regarding prompting/orchestration choices, the key design decisions (blind ranking, 4-round protocol) are validated in the ablation study (Table 3). Our prompt templates follow standard instruction formats and are provided in Appendix I for full transparency.

---

> > ### Author Rebuttal · Reviewer_6z7W · 2026-04-03
> >
> > The authors have addressed my questions. I will raise the score accordingly.

---

> > > ### Author Response · Authors · 2026-04-04
> > >
> > > Dear Reviewer 6z7W,
> > >
> > > Thank you for taking the time to review our rebuttal and for acknowledging that your concerns have been addressed. We greatly appreciate your constructive feedback, which has helped improve our work.
> > >
> > > Best regards,
> > >
> > > The Authors

---

### Decision · Program_Chairs · 2026-04-30

**Decision:**

Accept (regular)

**Comment:**

The paper proposes meta-evaluation as a way to address LLM training on non-verifiable tasks. It introduces CoNL, a multi-agent self-play framework, where multiple agents propose, critique and revise solutions and are jointly optimized.

Strength:
* The paper is written clearly and well-motivated.
* The training protocol is well-structured.
* The results are promising.

Weakness:
* The experiments in the initial version are on verifiable tasks despite the motivation on non-verifiable tasks. The authors added more non-verifiable tasks during the rebuttal, which should be included and analyzed in more detailed in the final version to support the motivation.
* I agree with the reviewer that adding more baselines (simpler one and complicated ones) is important.
* The reviewers brought up constructive feedback and it'd be great if the authors can integrate the rebuttal in the final version.